# ALK ligand ALKAL2 potentiates MYCN-driven neuroblastoma in the absence of *ALK* mutation

Marcus Borenäs[1,†], Ganesh Umapathy[1,†], Wei-Yun Lai[1], Dan E Lind[1] ID, Barbara Witek[2], Jikui Guan[1,3], Patricia Mendoza-Garcia[1] ID, Tafheem Masudi[1], Arne Claeys[4], Tzu-Po Chuang[1], Abeer El Wakil[2,‡] ID, Badrul Arefin[1], Susanne Fransson[5] ID, Jan Koster[6], Mathias Johansson[7] ID, Jennie Gaarder[5], Jimmy Van den Eynden[4,*] ID, Bengt Hallberg[1,**] ID & Ruth H Palmer[1,***] ID

## Abstract

High-risk neuroblastoma (NB) is responsible for a disproportionate number of childhood deaths due to cancer. One indicator of high-risk NB is amplification of the neural *MYC* (*MYCN*) oncogene, which is currently therapeutically intractable. Identification of anaplastic lymphoma kinase (ALK) as an NB oncogene raised the possibility of using ALK tyrosine kinase inhibitors (TKIs) in treatment of patients with activating ALK mutations. 8–10% of primary NB patients are ALK-positive, a figure that increases in the relapsed population. ALK is activated by the ALKAL2 ligand located on chromosome 2p, along with *ALK* and *MYCN*, in the "2p-gain" region associated with NB. Dysregulation of ALK ligand in NB has not been addressed, although one of the first oncogenes described was *v-sis* that shares > 90% homology with PDGF. Therefore, we tested whether ALKAL2 ligand could potentiate NB progression in the absence of ALK mutation. We show that ALKAL2 overexpression in mice drives ALK TKI-sensitive NB in the absence of ALK mutation, suggesting that additional NB patients, such as those exhibiting 2p-gain, may benefit from ALK TKI-based therapeutic intervention.

**Keywords** 2p-gain; ALK; ALKAL; MYCN; neuroblastoma
**Subject Categories** Cancer; Signal Transduction
The EMBO Journal (2021) 40: e105784

## Introduction

Anaplastic lymphoma kinase (ALK) is a receptor tyrosine kinase that is activated by the ligands ALKAL1 (FAM150A/AUGβ) and ALKAL2 (FAM150B/AUGα) (Morris *et al*, 1994; Iwahara *et al*, 1997; Guan *et al*, 2015; Reshetnyak *et al*, 2015). Oncogenic ALK was initially described as a nucleophosmin (NPM)-ALK fusion in anaplastic large cell lymphoma (ALCL) (Morris *et al*, 1997). Many other ALK fusion proteins have since been described in different cancer forms, such as non-small-cell lung cancer, diffuse large B-cell lymphoma (DLBCL) and inflammatory myofibroblastic tumour (IMT) (Hallberg & Palmer, 2013; Umapathy *et al*, 2019). Aberrant activation of ALK has also been reported in the childhood cancer neuroblastoma (NB), where both germline and somatic point mutations, predominantly in the kinase domain of the receptor, have been reported (Caren *et al*, 2008; Chen *et al*, 2008; George *et al*, 2008; Janoueix-Lerosey *et al*, 2008; Mosse *et al*, 2008).

High-risk NB is notoriously difficult to treat and typically exhibits a low mutation load as many paediatric cancers (Brodeur, 2003; Maris *et al*, 2007; Pugh *et al*, 2013; Grobner *et al*, 2018; Ma *et al*, 2018). In contrast, chromosomal aberrations such as deletions of parts of chromosome arms 1p and 11q, 17q gain, triploidy, as well as *MYCN* and *ALK* amplifications, are important for prognosis in NB (Brodeur, 2003; Vandesompele *et al*, 2005; Michels *et al*, 2007; Caren *et al*, 2008, 2010; Janoueix-Lerosey *et al*, 2008; De Brouwer *et al*, 2010). One long accepted indicator of high-risk NB and poor prognosis is amplification of the currently therapeutically intractable *MYCN* oncogene on chromosome 2p24, which is observed in 20–30% of all NB cases (Schwab *et al*, 1984; Brodeur, 2003; Maris *et al*, 2007). The identification of *ALK* as an oncogene in both

1 Department of Medical Biochemistry and Cell Biology, Institute of Biomedicine, Sahlgrenska Academy, University of Gothenburg, Gothenburg, Sweden
2 Department of Molecular Biology, Umeå University, Umeå, Sweden
3 Children's Hospital Affiliated to Zhengzhou University, Zhengzhou, China
4 Department of Human Structure and Repair, Anatomy and Embryology Unit, Ghent University, Ghent, Belgium
5 Laboratory Medicine, Institute of Biomedicine, Sahlgrenska Academy, University of Gothenburg, Gothenburg, Sweden
6 Department of Oncogenomics, Academic Medical Center, University of Amsterdam, Amsterdam, The Netherlands
7 Clinical Genomics, Science for life laboratory, University of Gothenburg, Gothenburg, Sweden
 *Corresponding author. Tel: +32 9 3324855; E-mail: jimmy.vandeneynden@ugent.be
 **Corresponding author. Tel: +46 31 7863815; E-mail: bengt.hallberg@gu.se
 ***Corresponding author (lead contact). Tel: +46 31 7863906; E-mail: ruth.palmer@gu.se
 †These authors contributed equally to this work as first authors
 ‡Present address: Department of Biological Sciences, Alexandria University, Alexandria, Egypt

familial and somatic NB raised the possibility of using ALK tyrosine kinase inhibitors (TKIs) in the treatment of NB patients who harbour activating ALK mutations. Initial clinical results with the first-generation ALK TKI crizotinib were disappointing in spite of some responses (Mosse *et al*, 2013). However, a number of studies have since examined next-generation ALK TKIs such as ceritinib, lorlatinib, brigatinib, alectinib and repotrectinib in a preclinical setting, identifying more potent inhibitors for the ALK mutant variants found in NB (Guan *et al*, 2016; Infarinato *et al*, 2016; Iyer *et al*, 2016; Siaw *et al*, 2016; Guan *et al*, 2018; Alam *et al*, 2019; Cervantes-Madrid *et al*, 2019). While the number of ALK mutation-positive NB patients on primary diagnosis is in the range of 8–10%, this figure increases substantially in the relapsed patient population (Martinsson *et al*, 2011; Schleiermacher *et al*, 2014; Eleveld *et al*, 2015). Since considerable morbidity is associated with high-risk NB protocols, it is important to thoroughly explore and identify all NB patient populations that may benefit from clinical use of ALK TKIs.

Activation of ALK signalling by ALKAL ligands has been shown to be important in the developing zebrafish neural crest, the tissue from which NB arises (Guan *et al*, 2015; Reshetnyak *et al*, 2015; Mo *et al*, 2017; Fadeev *et al*, 2018). In the human genome, *ALKAL2* is located on the distal portion of chromosome 2 (at 2p25), along with *ALK* and *MYCN*, in the "2p gain" region that has been associated with NB (Jeison *et al*, 2010; Javanmardi *et al*, 2019). We know from previous work that ALK activation drives transcription of *MYCN* and potentiates MYCN-driven NB in mouse and zebrafish models (Weiss *et al*, 1997; Berry *et al*, 2012; Heukamp *et al*, 2012; Schonherr *et al*, 2012; Zhu *et al*, 2012; Cazes *et al*, 2014; Ono *et al*, 2019), and this is supported by analysis of NB tumours where coexistence of *ALK* activating mutations and *MYCN* amplification forms a high-risk NB group with poor prognosis (De Brouwer *et al*, 2010). While our advances in genetic profiling of tumours have led to significant advances, this approach does not address signalling activity in cancer cells that may not be reflected by genetic mutation (Yaffe, 2019). As illustration, it is unclear whether ALK activity in the absence of mutation drives NB progression, and this could hypothetically be achieved by misregulation of ALK ligands. Indeed, one of the first oncogenes described was *v-sis*, which causes glioblastoma in marmoset monkeys and shares more than 90% homology with the PDGFB ligand (Doolittle *et al*, 1983; Waterfield *et al*, 1983; Heldin *et al*, 2018). Since this finding, PDGF ligand dysregulation has been described in several human cancers, including glioblastoma and the rare skin tumour dermatofibrosarcoma protuberans (DFSP) where a chromosomal translocation event between *PDGFB* and *collagen 1A1* results in a tumour promoting PDGF-like protein (Heldin *et al*, 2018). Thus, our considerable body of knowledge regarding PDGF ligands and their receptors in tumorigenesis highlights a potential scenario, whereby aberrant regulation of ALK ligands may activate ALK signalling via autocrine or paracrine stimulation to promote NB development. Indeed, in support of this scenario it has been reported that many NB exhibits high ALK expression in the absence of mutation and that these high levels correlate with poor progression (Lamant *et al*, 2000; Osajima-Hakomori *et al*, 2005; Janoueix-Lerosey *et al*, 2008; Mosse *et al*, 2008; Passoni *et al*, 2009; Duijkers *et al*, 2012; Wang *et al*, 2013; Regairaz *et al*, 2016; Javanmardi *et al*, 2019).

In this work, we have tested the hypothesis that ALKAL ligand overexpression is able to drive NB progression in the absence of

ALK receptor mutation. We show by RNA-Seq, total proteomics and phosphoproteomics that ALKAL stimulation of NB cell lines results in an ALK signalling response that is sensitive to ALK TKIs. Having characterized the ALKAL2/ALK signalling response *in vitro*, we tested the hypothesis that ALKALs drive ALK signalling *in vivo*. For this, we employed the *Th-MYCN* mouse model in which overexpression of MYCN in the neural crest drives NB development in mice (Weiss *et al*, 1997). Critically, the appearance of NB in *Th-MYCN* mice displays (i) incomplete penetrance and (ii) late onset (Weiss *et al*, 1997). We show here that overexpression of ALKAL2 is sufficient to drive rapid onset and highly penetrant *Th-MYCN*-driven NB in the absence of *Alk* mutation. Remarkably, these *Alkal2;Th-MYCN*-driven NBs are similar to ALK gain-of-function-driven NB as assessed by RNA-Seq and moreover respond to ALK TKI treatment. Together, these results indicate that aberrant regulation of the ALKAL2 ligand can drive NB, and most importantly suggest that a proportion of "ALK mutation-negative" NB patients may also benefit from ALK TKI-based therapeutic intervention.

## Results

### ALKAL2 stimulates ALK downstream signalling in NB cells

We and others have previously shown that ALKAL2 stimulates ALK in NB cells (Guan *et al*, 2015; Reshetnyak *et al*, 2015). In addition, several studies have reported changes in ALK downstream signalling in response to ALK TKI treatment in NB cells (Emdal *et al*, 2018; Van den Eynden *et al*, 2018). We therefore performed a comprehensive analysis of NB cells stimulated with ALKAL2 ligand. For this analysis, we employed NB1 and IMR-32 cells, which both express a wild-type (WT) ALK receptor.

To verify whether ALK signalling was indeed induced by ALKAL2, we first stimulated NB1 and IMR-32 cells with ALKAL2 for 30 min and 24 h, and monitored ALK activation by immunoblotting against pY1278-ALK and downstream signalling with pAKT, pERK and pS6 (Appendix Fig S1). ALK signalling in response to ALKAL2 ligand stimulation was rapid, and both pY1278-ALK and stimulation of downstream molecules could be observed in NB1 cells at 30 min with some response remaining at 24 h (Appendix Fig S1). A much stronger response was noted in NB1 cells, likely due to amplification of *ALK* and increased ALK receptor expression in comparison to IMR-32 cells. As expected, addition of the ALK TKI lorlatinib led to a complete inhibition of ALKAL2-induced signals (Appendix Fig S1).

Having confirmed that ALKAL2 stimulation results in the activation of ALK signalling that is inhibited by ALK TKI treatment, we performed RNA-Seq, harvesting samples at 1, 6 and 24 h time points (Fig 1A, Table EV1). At 1 h, we noted 34 and 13 genes that were upregulated ($\log_2$FC > 2 at 1% FDR) in NB1 and IMR-32 cells, respectively (no downregulation was observed; Fig 1B and C). We identified a set of six transcription factors (*EGR1*, *EGR2*, *EGR3*, *ARC*, *FOS* and *FOSB*) that were upregulated in both cell lines, and whose upregulation was sensitive to lorlatinib, suggesting that these effects were mediated via the ALK receptor (Fig 1B and C). This transcriptional response to ALKAL2 stimulation was transient and was no longer observed at either 6- or 24-h time points (Fig 1D). In line with our findings, a downregulation of these genes was observed

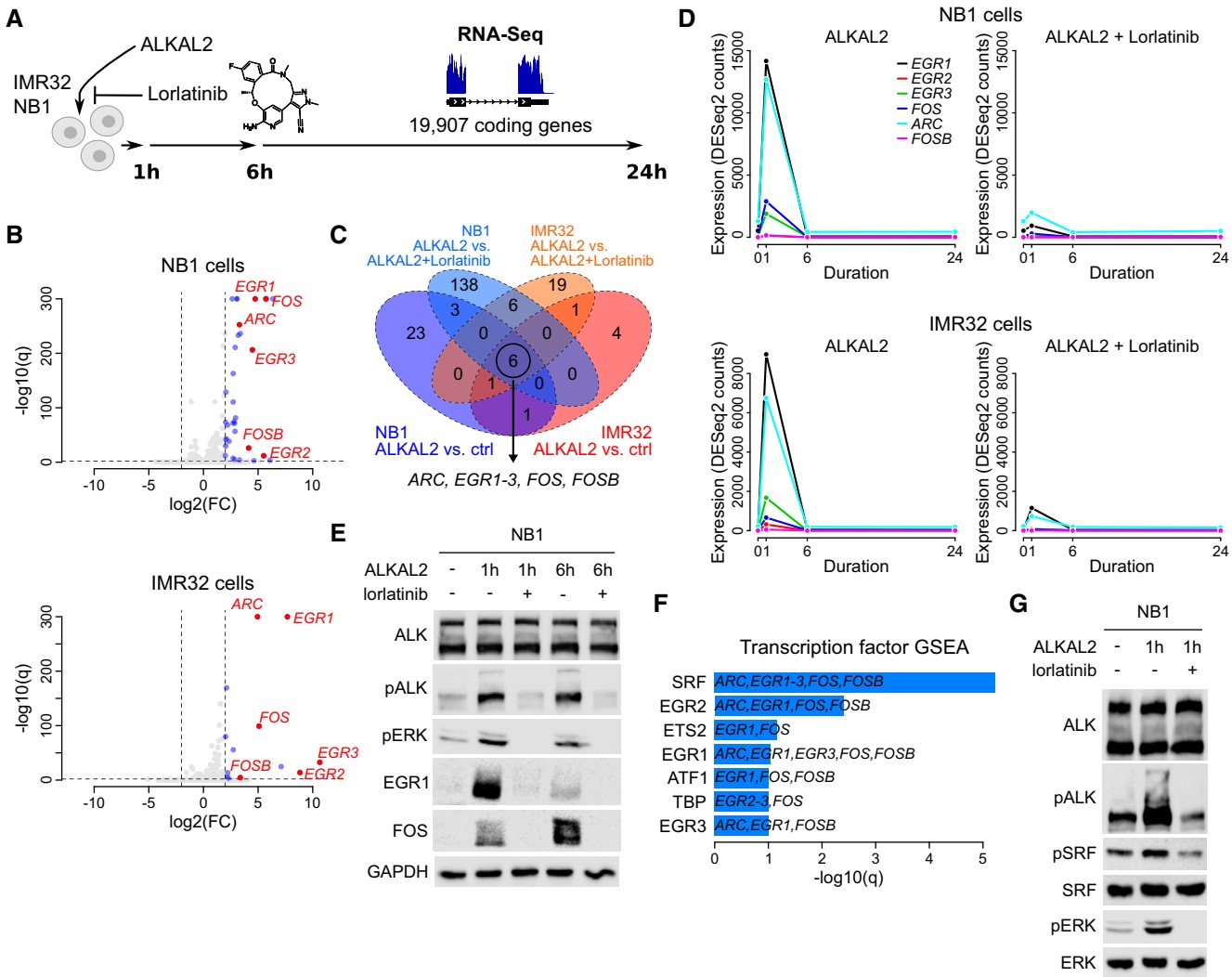

**Figure 1. ALKAL2 stimulates ALK signalling and transcriptional responses in NB cells.**

A RNA-Seq-based differential gene expression (DE) was measured in NB1 and IMR32 NB cell lines in response to ALKAL2 stimulation. See Table EV1 for detailed results.

B Volcano plot showing DE 1 h after NB1 (top) and IMR32 (bottom) cell treatment with ALKAL2. Dashed lines show DE thresholds. Up-/downregulated genes indicated in blue. Six genes that are DE in both cell lines and sensitive to the ALK inhibitor lorlatinib are indicated and labelled in red.

C Venn diagram indicating the number of DE genes between different conditions as indicated. Outer circles (labels below diagram) indicate the number of DE genes after ALKAL2 addition for NB1 cells (34 genes) and IMR32 cells (13 genes). Inner circles (labels on top) correspond to the number of DE genes after addition of lorlatinib. Six genes that are DE in both cell lines and sensitive to lorlatinib are indicated.

D Temporal dynamics of ALKAL2-induced transcription of *ARC*, *EGR1-3*, *FOS* and *FOSB* in NB1 and IMR32 cells in the presence and absence of lorlatinib, as indicated.

E Immunoblot validation of ALKAL2 induction of EGR1 and FOS at the protein level in NB1 cells. Cells were treated for 0, 1 and 6 h in the presence and absence of lorlatinib as indicated.

F Transcription factor prediction based on a gene set enrichment analysis (GSEA) of the identified six-gene set. Bar plot shows the $\log_{10}(q)$ values of all enriched transcription factors at 10% FDR.

G Immunoblot analysis of ALKAL2 induction of pSRF in NB1 cells. Cells were treated for 0 and 1 h in the presence and absence of lorlatinib as indicated.

Data information: RNA-Seq analysis was performed using three biological repeats. Immunoblots are representative of at least three independent experiments.

when ALK-driven cell lines CLB-BAR and CLB-GE were treated with lorlatinib, providing further support for the ALK specificity of this response ((Van den Eynden *et al*, 2018), Fig EV1). Additional protein validation experiments in both NB1 and IMR-32 cells confirmed the rapid lorlatinib-sensitive induction of EGR1 and FOS on ALKAL2 stimulation (Figs 1E and EV2). To predict the upstream transcription factors responsible for the observed expression

changes, we performed a gene set enrichment analysis (GSEA). The strongest enrichment was observed for serum response factor (SRF), which targets all 6 identified genes (Fig 1F). Because phosphorylation has been suggested as a mechanism of SRF activation in response to growth factor stimulation (Treisman, 1990), we hypothesized that ALKAL2 activates SRF through phosphorylation. We could indeed confirm a rapid and lorlatinib-sensitive appearance of

pSRF in NB cells stimulated with ALKAL2 (Figs 1G and EV2). We also examined *SRF* expression levels in NB patient tumours, employing the R2 database (http://r2.amc.nl). Investigation of two separate cohorts showed a trend of increased expression of *SRF* that correlated with poor prognosis in NB (Fig EV3); however, this does not take into account modulation of SRF activity at the post-transcriptional level.

Systematic characterization of ALK downstream signalling in NB cells based on a phosphoproteomic analysis has recently been reported (Emdal *et al*, 2018; Van den Eynden *et al*, 2018). To evaluate whether these signalling responses are similar after ALKAL2 induction, we stimulated NB1 cells in the absence or presence of lorlatinib and examined both the total proteomic (7,796 proteins) and phosphoproteomic (7,054 sites in 2,693 proteins) responses at 1 and 24 h (Fig 2A; Table EV2). Differential protein expression was most pronounced 24 h after ALKAL2 addition and sensitive to lorlatinib inhibition. Upregulation was observed for VGF, TNC, IGFBP5, FOSL2 and VIP (Fig 2B). Interestingly, apart from its upregulation after ALKAL2 stimulation, VGF was downregulated in response to lorlatinib, suggesting baseline ALK-dependent expression, consistent with earlier observations in NB1 cells (Emdal *et al*, 2018). In keeping with our proteomics dataset, we were able to verify induction of VGF protein at 24 h in NB1 cells that was abrogated on addition of lorlatinib (Fig 2C). Similarly, VGF protein levels in the ALK-driven CLB-BAR and CLB-GE cell lines were decreased in the presence of lorlatinib (Fig 2D). In parallel, we also performed a phosphoproteomic analysis which identified 80 phosphorylated and 40 dephosphorylated sites in response to 1 h ALKAL2 stimulation (Fig 2E). Among the most prominent phosphorylated targets, we found ALK, STAT3, CRK, FOXO3, RAB13, EIF1B and RPS6KC1, which have been reported to be modulated at the level of phosphorylation in response to ALK pathway inhibition in NB cells (Emdal *et al*, 2018; Van den Eynden *et al*, 2018). These differentially phosphorylated proteins were found to be enriched for several RTK pathways that have been related to ALK signalling such as NGF, FGFR, ERBB and AKT signalling pathways as well as neuronal development (Fig 2F, Appendix Fig S2). The phosphorylation response was ALK-dependent, as suggested by lorlatinib-sensitivity and, remarkably, very similar to the dephosphorylation response observed after ALK inhibition in NB1 or CLB-BAR cells (Fig EV4) (Emdal *et al*, 2018; Van den Eynden *et al*, 2018). Interestingly, in addition to the modulation of FOXO3 phosphorylation we also noted a lorlatinib-sensitive decrease in FOXO3 protein levels in response to ALK activation by ALKAL2 that could be seen at 24 h after ALKAL2 stimulation in NB1 cells (Fig 2G and H), highlighting the complex protein dynamics involved. One of the most prominent tyrosine phosphorylated targets in response to ALKAL2 stimulation was Y705 on STAT3. Phosphorylation of STAT3 was induced at 1 h and remained highly phosphorylated at 24 h after addition of ALKAL2 (Fig 2I). In the presence of lorlatinib, pY705-STAT3 was not detected (Fig 2I).

### *Alk-F1178S* mice are viable and exhibit sympathetic ganglion hyperplasia

Previous reports have shown that ALK collaborates with MYCN to drive NB in mouse models (Berry *et al*, 2012; Heukamp *et al*, 2012; Zhu *et al*, 2012; Cazes *et al*, 2014). Mutation of human *ALK-F1174*

in the ALK kinase domain, a hot spot in human NB, to either L/S/I/C or V, has been described as an aggressive mutation that is observed predominantly in sporadic NB cases (Hallberg & Palmer, 2013). The *ALK-F1174S* mutation was first described in a relapsed NB patient where ALK mutation correlated with aggressive disease progression (Martinsson *et al*, 2011). We generated an *Alk-F1178S* mouse by homologous recombination, leading to point mutation of residue F1178 of mouse ALK, a sequence equivalent to F1174 in human ALK (Appendix Fig S3). This results in an activated *Alk-F1178S* RTK under the control of physiological transcriptional regulation elements at the *Alk* locus. *Alk-F1178S* homozygous mice were obtained with expected Mendelian ratios, and a colony of *Alk-F1178S* mice was established. Since previous reports of Alk gain-of-function mice have reported hyperplasia in the sympathetic ganglia (Cazes *et al*, 2014; Ono *et al*, 2019), we investigated ganglia from *Alk-F1178S* mice and WT siblings at developmental stage P9. *Alk-F1178S* heterozygous caeliac ganglia were significantly enlarged and displayed hyperplasia when compared with controls, which was enhanced in *Alk-F1178S* homozygous animals (Fig 3A–C). Neither homozygous ($n = 161$) or heterozygous ($n = 416$) *Alk-F1178S* animals exhibited spontaneous tumours at birth. Further observation of heterozygous ($n = 13 > 18$ months) and homozygous ($n = 19 > 18$ months) *Alk-F1178S* animals up to 18 months of age did not reveal development of NB or any other type of cancer. Thus, while no gross tumour development is observed in *Alk-F1178S* mice, significant hyperplasia can be detected in Alk expressing neural crest-derived structures during development, such as the sympathetic ganglia, which is in agreement with other reports (Cazes *et al*, 2014; Ono *et al*, 2019).

### *Alk-F1178S* collaborates with *Th-MYCN* to drive neuroblastoma

*Alk-F1178S* animals were bred with *Th-MYCN* transgenic mice expressing MYCN under the control of the tyrosine hydroxylase (*Th*) promoter, and tumour development was followed. No gross tumour development was observed in heterozygous *Alk-F1178S* mice (Fig 3D). As previously reported, hemizygote *Th-MYCN* mice developed NB presenting as stroma-poor unilateral, single paraspinous masses enriched in small blue round cells in approximately 50% of mice at 40 wk (Fig 3D and E). Combining *Alk-F1178S* with *Th-MYCN* resulted in a significant increase in the aggressiveness of tumour development, exemplified by complete tumour penetrance at 30 wk together with markedly earlier tumour onset, averaging 8 wk (Fig 3D and E). Histologically, *Alk-F1178S;Th-MYCN* tumours were similar to those observed in *Th-MYCN* mice, although they were generally less bloody. We also examined sympathetic ganglion morphology in *Alk-F1178S*, *Th-MYCN*, *Alk-F1178S;Th-MYCN* mice and *WT* siblings at P9. Hyperplasia was observed in *Alk-F1178S* caeliac ganglia and was further increased in the presence of *Th-MYCN* (Fig 3F and G). Using Ki67 as a marker for cell proliferation, we noted significantly increased reactivity in *Alk-F1178S; Th-MYCN*, compared with *Th-MYCN* tumours, suggesting that the increased potential for tumour development is initiated at early stages in the sympathetic ganglia of animals bearing both oncogenes (Fig 3F and G). Therefore, in agreement with previous reports, we conclude that the activation of Alk under the control of its endogenous regulatory elements potentiates *Th-MYCN*-driven NB development.

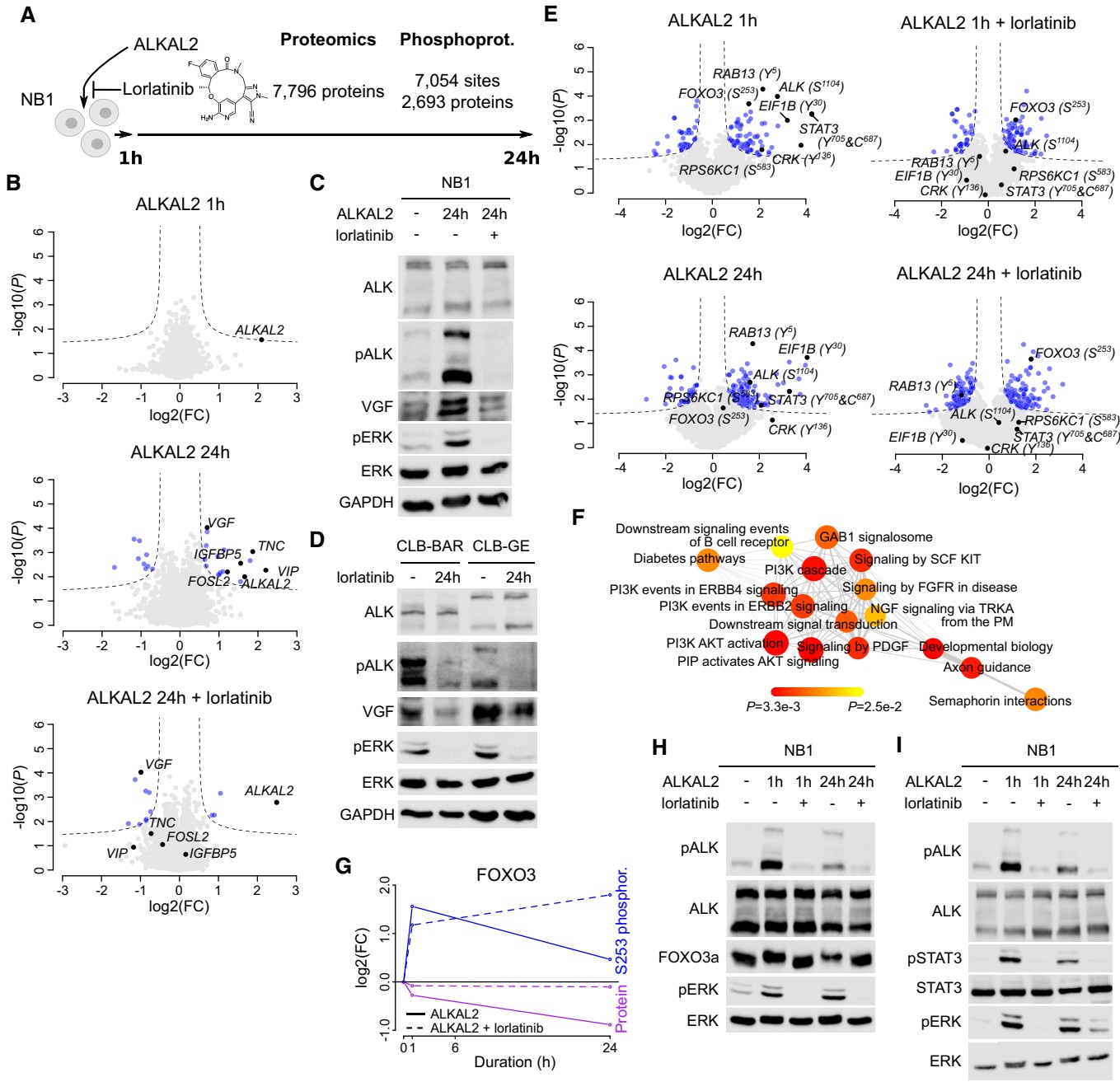

**Figure 2. ALKAL2 stimulation of downstream ALK signalling in NB cells at the post-transcriptional level.**

A   Differential protein expression and phosphorylation was determined in NB1 cells in response to ALKAL2 stimulation. See Table EV2 for detailed results.

B   Volcano plots showing differential protein expression 1 and 24 h after ALKAL2 in the presence or absence of lorlatinib stimulation as indicated. Dashed lines indicate differential expression thresholds. Differentially expressed proteins indicated in blue. Most pronounced responding proteins indicated in black and labelled.

C, D   Immunoblot analysis of VGF protein in NB cells. (C) NB1 cells after 24 h stimulation with ALKAL2 in the presence or absence of lorlatinib. (D) CLB-BAR and CLB-GE cells after 24 h inhibition with lorlatinib.

E   Volcano plots showing differential phosphorylation. Labelling colours as in (B).

F   GSEA network graph with nodes representing the enriched reactome pathways (at 25% FDR). Node sizes correlate to the normalized enrichment scores, node colours indicate *P* values (as in colour legend), and edge widths correspond to the number of overlapping genes between the connected nodes.

G   Graphical representation of FOXO3 dynamics, indicating S253 phosphorylation and total FOXO3 protein levels in response to ALKAL2 stimulation, in the presence or absence of lorlatinib.

H, I   Immunoblot validation of FOXO3a and STAT3 in response to ALKAL2 stimulation in the presence or absence of lorlatinib as indicated. The slower migrating FOXO3a band in SDS–PAGE in (H) likely reflects FOXO3a phosphorylation that is not seen in the presence of lorlatinib.

Data information: Proteomic analysis was performed using three biological repeats. Immunoblots are representative of at least three independent experiments.

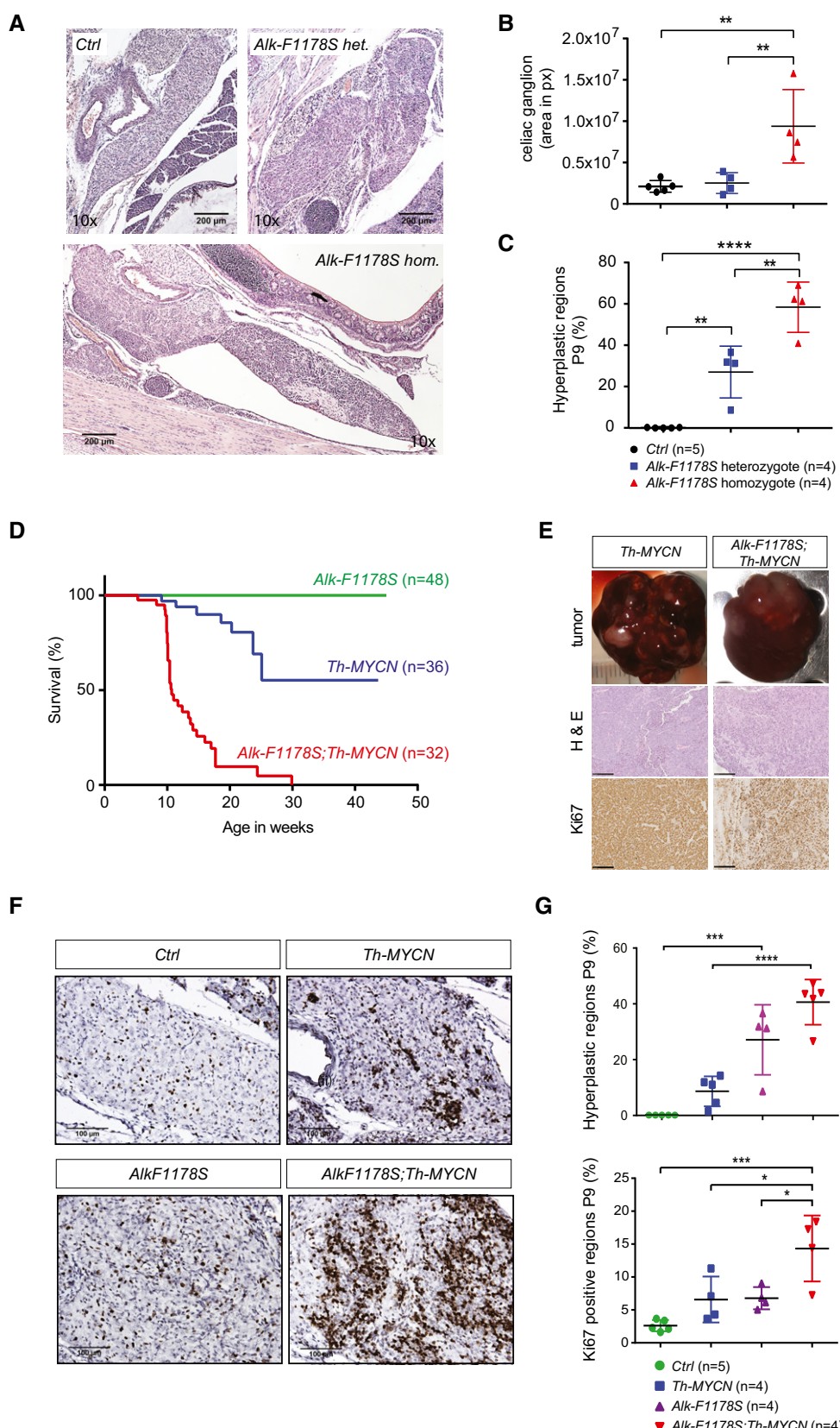

**Figure 3.**

◀

**Figure 3. *Alk-F1178S* collaborates with *Th-MYCN* to drive NB in mouse models.**

A   Haematoxylin and eosin staining of longitudinal sections of P9 pups at the central part of the left caeliac ganglion.
B   Quantification of the area of caeliac ganglia cross sections. Largest sections from the central part of left caeliac ganglions of different individuals were chosen for the analysis. **$P < 0.01$, one-way ANOVA followed by Tukey multiple comparison test. Data shown represent mean ± SD.
C   Hyperplasia quantification in central sections of P9 left caeliac ganglions shown as a per cent of hyperplastic regions areas per ganglion cross section. **$P < 0.01$, ****$P < 0.0001$, one-way ANOVA followed by Tukey multiple comparison test. Data shown represent mean ± SD.
D   Kaplan–Meier survival curve of mice resulting from intercrosses of *Th-MYCN* hemizygotes and *Alk-F1178S* heterozygote mice ($P < 0.0001$; log-rank test). Wild-type littermates were excluded.
E   Gross appearance, haematoxylin and eosin as well as Ki67-stained sections of representative *Th-MYCN* and *Alk-F1178S;Th-MYCN* tumours. Scale bars indicate 250 μm.
F   Ki67 immunohistochemical staining of P9 caeliac ganglia in mice of the indicated genotype (quantified in G).
G   Quantification of hyperplastic areas (shown as per cent of hyperplastic regions per ganglion cross section) and Ki67 expression (shown as positive for Ki67 staining areas per total area of the section through the ganglion central part at P9). (*$P < 0.05$, ***$P < 0.001$, ****$P < 0.0001$, one-way ANOVA followed by Tukey multiple comparison test. Data shown represent mean ± SD).

## Overexpression of ALKAL2 potentiates MYCN oncogenic activity *in vivo*

Since ALKAL stimulation of human NB cells results in a similar modulation of downstream signalling, as observed in ALK gain-of-function cells treated with ALK TKIs, we asked whether ALKAL ligands were able to drive NB development in mouse models. We first confirmed that the mouse ALKAL2 ligand was able to activate the mouse ALK RTK, as has previously been shown for human ALKAL2 and ALK (Guan *et al*, 2015; Reshetnyak *et al*, 2015). In both cell culture and an exogenous *Drosophila* fly eye assay, mouse ALKAL2 was able to robustly activate both the human and mouse ALK RTKs (Fig EV5). Next we generated transgenic mice expressing ALKAL2 (*Rosa26_Alkal2*; Appendix Fig S4). *Rosa26_Alkal2* homozygous mice were obtained with expected Mendelian ratios, and a colony was established. Similar to *Alk-F1178S*, no gross tumour development was observed in mice carrying the *Rosa26_Alkal2* transgene alone. *Rosa26_Alkal2* mice were bred with *Th-MYCN* transgenic mice and progeny monitored for tumour development. As expected from our previous results, *Alk-F1178S; Th-MYCN* mice displayed highly penetrant NB and rapid lethality (median survival 8.4 wk) when compared to *Th-MYCN* mice (Fig 4 A). Strikingly, mice heterozygote for *Rosa26_Alkal2* and *Th-MYCN* also showed a high tumour penetrance as well as a rapid lethality (Fig 4A) and a median survival of 10.1 wk even though they have a WT ALK receptor. No *Rosa26_Alkal2*, *Alk-F1178S* or WT mice developed tumours, and all remained healthy throughout the 200-day study. Tumours arising in *Rosa26_Alkal2;Th-MYCN* were indistinguishable in their presentation from those arising in *Th-MYCN* and *Alk-F1178S;Th-MYCN* animals. They appeared to originate primarily in the abdominal paraspinal ganglia, developing as locally invasive abdominal masses that only occasionally involved the adrenal glands (Fig 4C). 10% of *Rosa26_Alkal2;Th-MYCN* tumours ($n = 20$ examined) and 30% of *Alk-F1178S;Th-MYCN* ($n = 10$ examined) exhibited involvement of one or more adrenal gland. Histological and immunoblot analysis revealed small round blue cell tumours poor in stroma that expressed NCAM1, synaptophysin (SYP), Chromogranin A (CGA) and MYCN in *Rosa26_Alkal2;Th-MYCN* tumours, in agreement with NB (Fig 4D and E). Tumours also expressed ALKAL2 protein (Fig 4E), in keeping with our previous findings of ALKAL2 protein in human NB cells (Javanmardi *et al*, 2019). Careful monitoring of our mouse colony over time identified five out of 45 *Rosa26_Alkal2;Th-MYCN* animals that did not develop tumours at 200 days, prompting us to carefully review

tumour occurrence in all genotypes. Taken together, analysis of *Th-MYCN*, *Alk-F1178S;Th-MYCN* and *Rosa26_Alkal2;Th-MYCN* mice revealed a high level of tumour penetrance in both *Alk-F1178S;Th-MYCN* (98% at 200 days) and *Rosa26_Alkal2;Th-MYCN* (89% at 200 days), relative to that observed in *Th-MYCN* mice (46% at 200 days; Fig 4F). Moreover, estimated median survival for *Rosa26_Alkal2;Th-MYCN* and *Alk-F1178S;Th-MYCN* animals was similar, 10 and 10.4 wks, respectively, compared with that of *Th-MYCN* alone (median survival not reached at 200 days; Fig 4F).

Since human NB often exhibits chromosomal aberrations, we investigated genomic DNA of the various NB arising in our mice by whole genome sequencing (WGS). In general, there was a low mutational burden and lack of larger recurrent or syntenic copy number alterations. In total, 55 SNVs were detected among the eight tumours analysed with an average of 6.9 SNV per tumour (range 1–12; Table EV3). No gene was affected by recurrent SNVs in multiple samples, and no mutation was detected in well-established cancer genes. All 8 murine tumours had overall flat copy number alteration profiles, lacking larger segmental copy number aberrations and numerical alterations (Appendix Fig S5). No alterations associated with *Tert* or *Atrx*, nor alterations of areas syntenic to human chromosomal regions 17q, 11q or 1p were observed. However, 30 smaller focal deletions or gains were detected (Table EV3), with recurrent alterations affecting thee different genomic loci. These included deletions of *Tcf4*, *Macrod2* and a region distal to *Tenm3* (Appendix Fig S6).

Taken together, our data show that ALKAL2 ligand overexpression is able to collaborate with MYCN to drive NB in the absence of activating ALK mutations.

## ALKAL2-induced NB exhibits a transcriptional signature similar to that of activated ALK

ALKAL2-induced tumours were further investigated by RNA-Seq analysis. Tumour samples from *Th-MYCN*, *Alk-F1178S;Th-MYCN* and *Rosa26_Alkal2;Th-MYCN* mice were harvested and RNA-Seq data compared. The expression of the codon-optimized *Alkal2* transgene was first confirmed in *Rosa26_Alkal2;Th-MYCN* tumours (median 1.01 (0.79–1.07) reads per million versus 0 in the other 2 tumour types; Fig 5A). The expression of *Chga*, *Th*, *Dbh*, *Syp*, *Ncam1* and *Alkal2* was also observed in the tumour transcriptome of all three genotypes (Table EV4), which is in agreement with our histological analysis (Fig 4D). Tumour identity was further investigated by comparison of the overall gene expression signature with 6

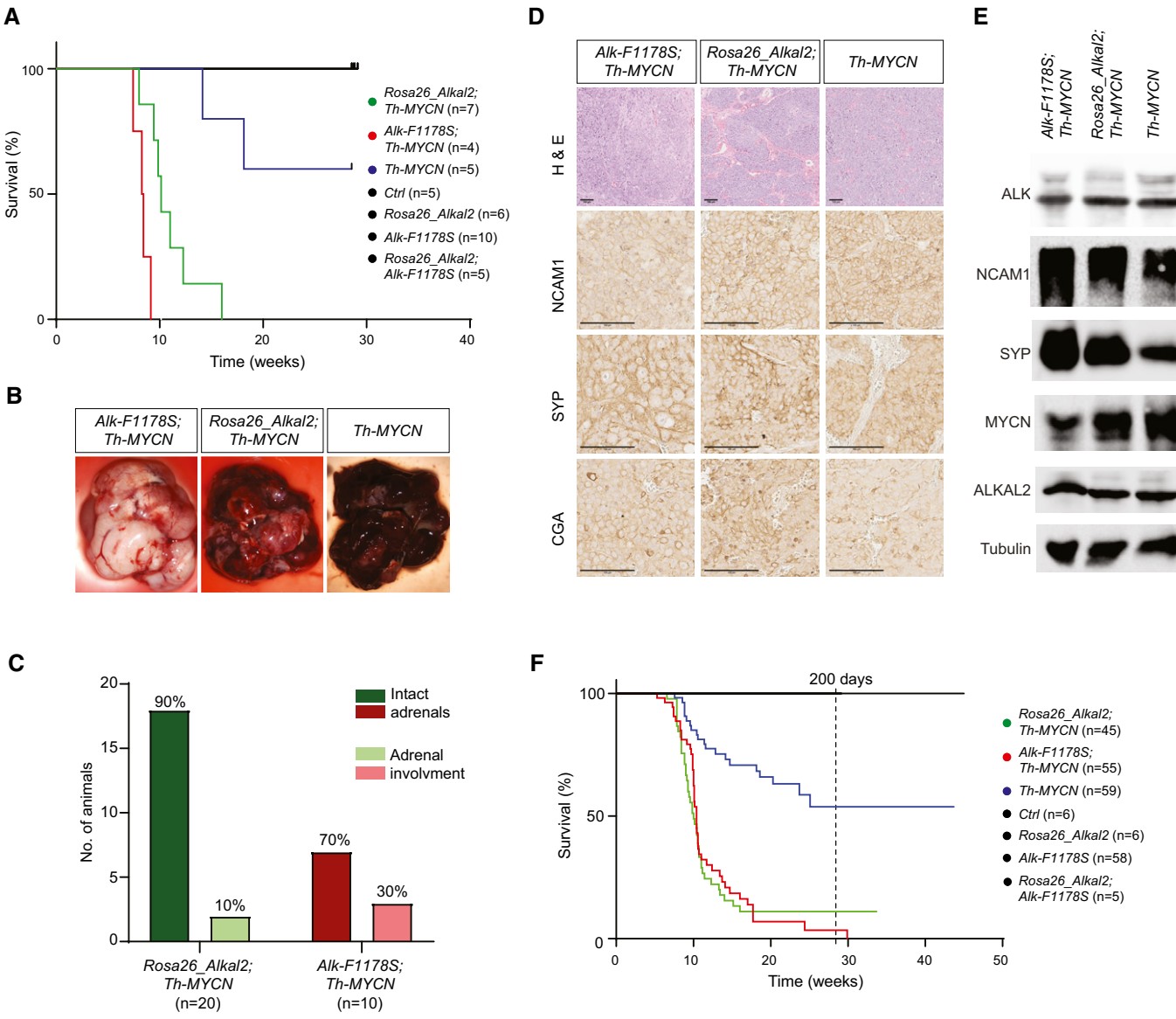

**Figure 4. ALKAL2 collaborates with MYCN to drive NB in mouse models.**

A   The oncogenic activity of MYCN is potentiated by overexpression of ALKAL2. Kaplan–Meier survival curves for *Rosa26_Alkal2;Th-MYCN*, *Alk-F1178S;Th-MYCN* and *Th-MYCN* mice. Also shown are *Rosa26_Alkal2N*, *Alk-F1178S* and control (Ctrl) mice. Comparison of survival of *Th-MYCN* alone and *Rosa26_Alkal2;Th-MYCN* curves showed a significant difference ($P = 0.003$; log-rank test).

B–E Tumours harvested from *Rosa26_Alkal2;Th-MYCN*, *Alk-F1178S;Th-MYCN* and *Th-MYCN* mice express NB markers. Tumours from all three genotypes were large, in most cases filling the abdominal cavity (B). Dissection post-mortem revealed that the majority of *Rosa26_Alkal2;Th-MYCN* (18/20) and *Alk-F1178S;Th-MYCN* (7/10) tumours did not involve the adrenal glands (C). Histological examination of *Rosa26_Alkal2;Th-MYCN*, *Alk-F1178S;Th-MYCN* and *Th-MYCN* tumours revealed positive staining for NCAM1, synaptophysin (SYP) and Chromogranin A (CGA) (D) that was confirmed for NCAM1 and SYP along with MYCN, ALK and ALKAL2 by immunoblotting (E). Scale bars indicate 100 μm. Immunoblots are representative of three independent technical analyses.

F   Accumulated Kaplan–Meier survival curves are shown for all monitored *Rosa26_Alkal2;Th-MYCN*, *Alk-F1178S;Th-MYCN* and *Th-MYCN* mice over time, estimating tumour penetrance ($P < 0.001$; log-rank test).

common human cancer types using a principal component analysis, revealing the highest similarity with human NB tumours, underlining the validity of our mouse model (Fig 5B). We then compared the transcriptional effects of *Rosa26_Alkal2;Th-MYCN* with *Th-MYCN* and identified 23 upregulated and 17 downregulated genes (log$_2$FC threshold 2 at 1% FDR; Fig 5C and Table EV4). While this number of responding genes was an order of magnitude lower as

compared to the *Alk-F1178S;Th-MYCN* tumours (381 differentially expressed genes), 52.5% (21/40) differentially expressed genes in *Rosa26_Alkal2;Th-MYCN* overlapped with the response in *Alk-F1178S;Th-MYCN* tumours ($P = 6.9e-27$, Fisher's exact test; Fig 5D). In general, the transcriptional signature of *Rosa26_Alkal2;Th-MYCN* tumours was less pronounced but overall very similar to the signature in *Alk-F1178S;Th-MYCN* tumours (Fig 5E). We also noted

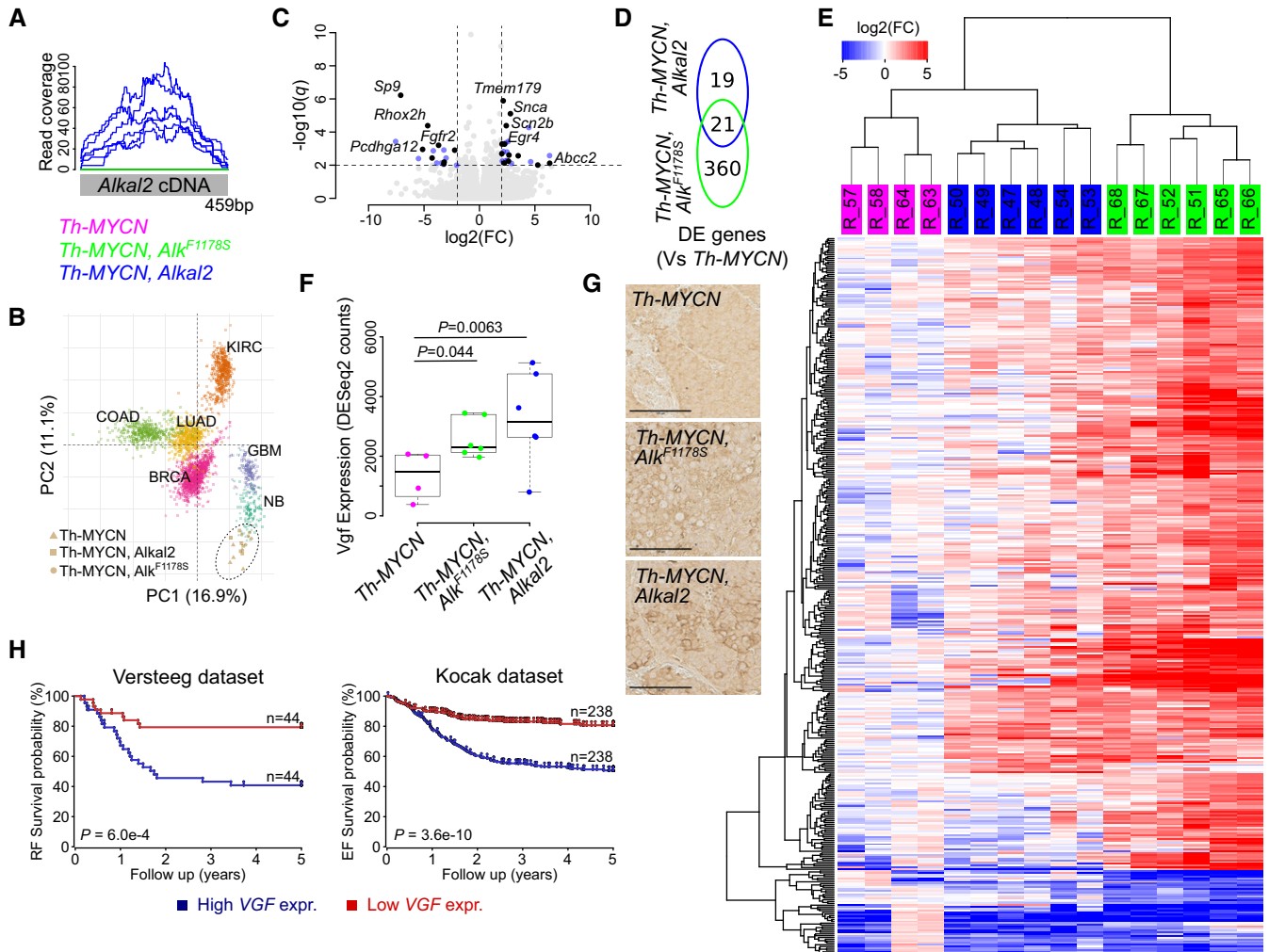

**Figure 5. ALKAL2-driven tumours share a transcriptional signature with ALK-F1178S-driven NB.**

RNA-Seq-based differential gene expression analysis of tumours arising in *Rosa26_Alkal2;Th-MYCN* (*Alkal2*) [*n* = 6], *Alk-F1178S;Th-MYCN* (*Alk^F1174S^*) [*n* = 6] and *Th-MYCN* (*MYCN*) mice [*n* = 4]. See Table EV4 for detailed results.

A Read coverage of the codon-optimized *Alkal2* transgene, confirming *Alkal2* expression in *Alkal2* tumours.

B Principal component (PC) analysis of the expression signature of human neuroblastoma (NB) and five other human cancers (BRCA: breast adenocarcinoma; COAD: colon adenocarcinoma; LUAD: lung adenocarcinoma; KIRC: kidney renal clear cell carcinoma; GBM: glioblastoma multiforme) with mice tumour samples mapped independently using PC coordinates. *MYCN* amplified NB samples are indicated by circles, and non-amplified samples are indicated by squares.

C, D Volcano plot showing differential expression (DE) between *Alkal2* and *MYCN* tumours. Differentially expressed genes are shown in blue (DE in *Alkal2* tumours only) or black (DE in both *Alkal2* and *Alk^F1174S^* tumours, as shown in [D]). Top ranked genes labelled. Dashed lines represent DE cut-offs.

E DE heatmap based on unsupervised hierarchical clustering of 400 DE genes (rows) and 16 samples (columns, as indicated on top). Sample colour legend as in (A). Colour key shown on top left.

F Boxplot showing *Vgf* expression in the three tumour types as indicated. Box plots indicate median values and lower/upper quartiles with whiskers extending to 1.5 times the interquartile range. *P* values calculated using Wald test as reported by DESeq2.

G Histological examination of *Th-MYCN, Rosa26_Alkal2;Th-MYCN* and *Alk-F1178S;Th-MYCN* tumours revealing positive staining for VGF. Scale bars indicate 100 μm.

H Kaplan–Meier relapse-free (RF) and event-free (EF) survival probability curves from two different NB cohorts, the Versteeg 88 cohort (left panel) and the Kocak 649 cohort (right panel), as derived from the R2 platform. Patients with higher *VGF* expression are highlighted in blue, whereas patients with lower expression are highlighted in red. The log-rank test *P* values are indicated.

increased levels of VGF both in *Alk-F1178S;Th-MYCN* and *Rosa26_Alkal2;Th-MYCN* tumours (Fig 5F and G), in agreement with our earlier observation of strongly upregulated VGF protein levels in NB1 cells stimulated with ALKAL2 (Fig 2, Table EV2) and a previous report of increased *Vgf* mRNA levels in an ALK gain-of-function NB mouse model (Cazes *et al*, 2014). Since VGF has

recently been reported to promote survival and growth of glioblastoma cells (Wang *et al*, 2018), we examined VGF expression levels in NB patient tumours, employing the R2 database (http://r2.amc.nl). Investigation of two separate cohorts showed a correlation of increased expression of *VGF* with poor relapse-free (RF) and event-free (EF) survival probability NB (Fig 5H). We also noted a

significant correlation of high *VGF* with poor prognosis in terms of overall survival (log-rank test *P* = 3.3e-16 for the Kocak cohort and log-rank test *P* = 5.2e-05 for the Versteeg cohort).

### ALKAL2-driven tumour-derived NB cell lines respond to ALK TKI treatment

To investigate whether ALKAL2-driven NB is sensitive to ALK TKI treatment, we first established murine NB cell lines from tumours harvested from *Rosa26_Alkal2;Th-MYCN* and *Alk-F1178S;Th-MYCN* mice. Cell line #3456 was generated from an *Alk-F1178S;Th-MYCN* NB, while cell line #3540 was generated from *Rosa26_Alkal2;Th-MYCN* tumour tissue. Increased levels of ALKAL2 protein expression were confirmed in the *Rosa26_Alkal2;Th-MYCN* derived #3540 cell line (Fig 6A). To investigate their response towards ALK TKI treatment, cells were treated with brigatinib (a second-generation ALK TKI) and their growth monitored. Addition of brigatinib resulted in a significant growth suppression on these newly generated NB cell

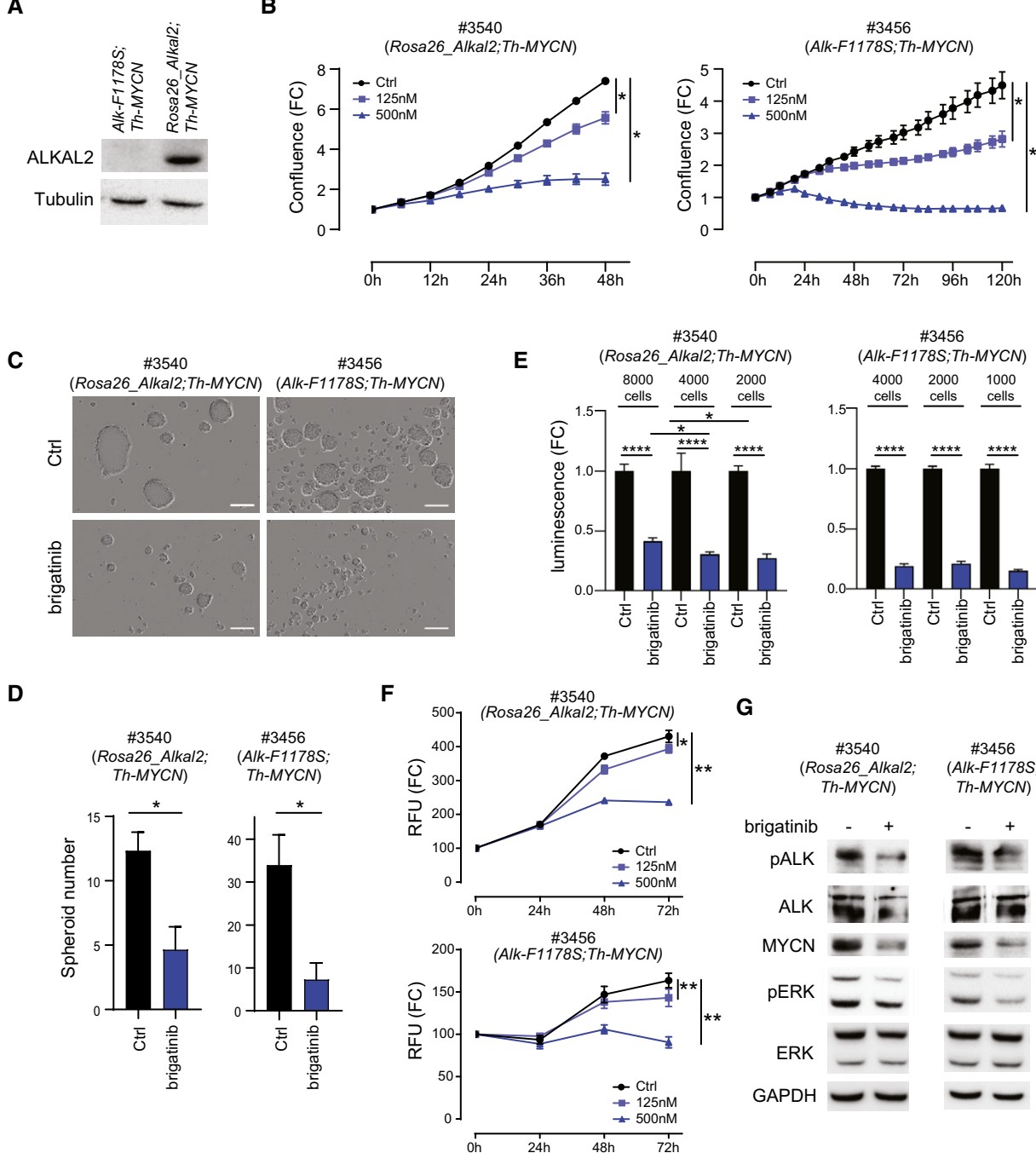

**Figure 6.**

**Figure 6. Cell lines derived from ALKAL2-driven NB respond to ALK TKI treatment.**

Murine NB cell lines were generated from tumours arising in *Rosa26_Alkal2;Th-MYCN* (#3540) and *Alk-F1178S;Th-MYCN* (#3456) mice.

A   Alkal2 expression in cells derived from *Rosa26_Alkal2;Th-MYCN* (#3540) and *Alk-F1178S;Th-MYCN* (#3456) NB. Immunoblotting analysis for ALKAL2 and tubulin in the indicated mouse NB cell lines. Whole cell lysates (30 µg) were analysed in each lane.

B   The effect of increasing concentrations of brigatinib on cell confluence was analysed by IncuCyte Live Cell Analysis of both *Rosa26_Alkal2;Th-MYCN* (#3540) and *Alk-F1178S;Th-MYCN* (#3456) cell lines. Data are presented as mean ± SEM from three independent experiments. *$P < 0.05$, **$P < 0.005$; two-tailed paired Student's *t*-test.

C, D   Brigatinib suppressed tumour spheroid formation and spheroid viability. Cells (#3456 or #3540) were treated with brigatinib (0, 150 nM) for 4 days in ultra-low attachment plates. The spheroid number was analysed by IncuCyte Live Cell Analysis. Data are presented as means ± SEM from three independent experiments. *$P < 0.05$, two-tailed unpaired Student's *t*-test. Scale bar (C) is 200 µm.

E   Tumour spheroids formed from either *Rosa26_Alkal2;Th-MYCN* (#3540) or *Alk-F1178S;Th-MYCN* (#3456) were formed at indicated cell number in ultra-low attachment plates for 3 days and followed by brigatinib (0 or 150 nM) for 10 days. Inhibitor was re-fed every other day. Cell viability was determined by CellTiter-Glo 3D cell viability kit and data are presented as mean ± SEM from five independent experiments. *$P < 0.05$, ****$P < 0.0001$, two-tailed unpaired Student's *t*-test.

F   Cell viability. Mouse tumour-derived cell lines #3540 (*Rosa26_Alkal2;Th-MYCN*) and #3456 (*Alk-F1178S;Th-MYCN*) were treated with brigatinib (125 or 500 nM), and viability was evaluated by using a resazurin-based assay. Data are presented as mean ± SEM from three independent experiments. *$P < 0.05$, **$P < 0.005$; two-tailed paired student *t*-test.

G   Brigatinib treatment (0 or 150 nM) for 6 h resulted in inhibition of ALK phosphorylation, and of activation of downstream signalling (ERK1/2), as well as MYCN expression. Cell lysates (*Rosa26_Alkal2;Th-MYCN* (#3540) and *Alk-F1178S;Th-MYCN* (#3456)) were immunoblotted with the indicated antibodies.

lines in a dose-dependent manner (Fig 6B). In order to more closely mimic the NB tissue and microenvironment, the effect of brigatinib was tested on both spheroid formation ability and viability in 3D tumour spheroid cultures. Brigatinib significantly inhibited spheroid formation and viability in both NB cell lines (Fig 6C–F). We also observed that smaller *Rosa26_Alkal2;Th-MYCN* spheroids were more sensitive to brigatinib treatment when compared to larger ones (Fig 6E). The effect of ALK inhibition on downstream signalling pathways was determined by immunoblotting, with decreased phosphorylation levels of ALK, downstream signalling (phospho-ERK1/2), as well as MYCN expression after 6 h of ALK TKI treatment (Fig 6G). In sum, both murine NB cell lines, harbouring either the ALK-F1178S gain-of-function mutation (*Alk-F1178S;Th-MYCN*) or with ALKAL2 ligand overexpression (*Rosa26_Alkal2;Th-MYCN*), were sensitive to ALK inhibition, suggesting that ALKAL2-driven NB may respond to ALK TKI treatment.

### ALKAL2-driven NB responds to ALK TKI treatment

As *Rosa26_Alkal2;Th-MYCN* tumour-derived NB cells are sensitive to ALK TKI inhibition, we tested whether NB tumour development could be inhibited in mice. We have previously shown that tumour growth of *Th-ALK-F1174L;Th-MYCN*-driven NB is inhibited by treatment with lorlatinib (Guan *et al*, 2016). Cells dissociated from NB tumour tissue arising from either *Rosa26_Alkal2;Th-MYCN* and *Alk-F1178S;Th-MYCN* were subjected to increasing doses of either brigatinib or lorlatinib. Tumour cells of both genotypes displayed dose-dependent sensitivity to lorlatinib as well as brigatinib (Fig 7A). To test whether ALKAL2-driven NB was sensitive to ALK TKI treatment *in vivo*, we treated NB tumours arising in *Rosa26_Alkal2;Th-MYCN* mice with lorlatinib (10 mg/kg body weight, 2× per day) for a period of 14 days and monitored tumour growth by ultrasound. Tumour growth was significantly inhibited in the lorlatinib-treated group as compared to controls (Fig 7B). No significant weight loss was observed in the lorlatinib-treated group (Fig 7C). Both ultrasound and MRI analyses allowed visualization of highly aggressive rapidly growing NB in *Rosa26_Alkal2;Th-MYCN* animals that within 14 days filled the abdominal cavity (Fig 7D and E). This can be compared with the restricted growth of NB tumours in *Rosa26_Alkal2;Th-MYCN* when

treated with lorlatinib (Fig 7D and E). Histological analysis of lorlatinib-treated tumours further supported a reduced rate of growth, with a significant decrease in phospho-histone H3 (pH3)-positive cells in treated tumours when compared with controls (Fig 7F). These data indicate that ALKAL2-driven NB is sensitive to ALK TKI treatment.

## Discussion

Our appreciation of the importance of developmental processes in NB tumorigenesis has increased over the last decade. One of the best studied NB models is the *Th-MYCN* mouse. This model exhibits late onset and variable penetrance, dependent on genetic background (Weiss *et al*, 1997). A number of groups have shown that ALK collaborates with MYCN to drive NB when overexpressed in mice (Berry *et al*, 2012; Heukamp *et al*, 2012). The first report of an ALK GOF mouse knock-in showed that a single point mutation in the ALK kinase domain was sufficient to drive NB in collaboration with MYCN overexpression (Cazes *et al*, 2014). Our findings here confirm that mice harbouring ALK GOF knock-in (in this case F1178S, corresponding to human F1174S (Martinsson *et al*, 2011)) also exhibit enlarged sympathetic ganglia and drive NB in collaboration with MYCN.

Given the strong body of evidence implicating ALK activation in NB development, it is also important to address the potential role of ligands for this RTK. While much attention to date has focused on identification of ALK activating mutations, overexpression and activation of ALK in the absence of kinase domain mutations has also been reported (Janoueix-Lerosey *et al*, 2008; Mosse *et al*, 2008; Duijkers *et al*, 2012; Chang *et al*, 2020). Tumour development driven by misregulation of receptor ligands is an important consideration in NB, underscored by the fact that one of the first oncogenes described was the v-sis oncogene that shares more than 90% homology with the PDGF ligand (Heldin *et al*, 2018).

Since the identification of the ALK ligands (Guan *et al*, 2015; Reshetnyak *et al*, 2015), the question of whether ALKAL misregulation has consequences in NB has remained unanswered. A role for ALKAL ligands in the development of the vertebrate neural crest, the tissue from which NB arises, has been reported in the zebrafish

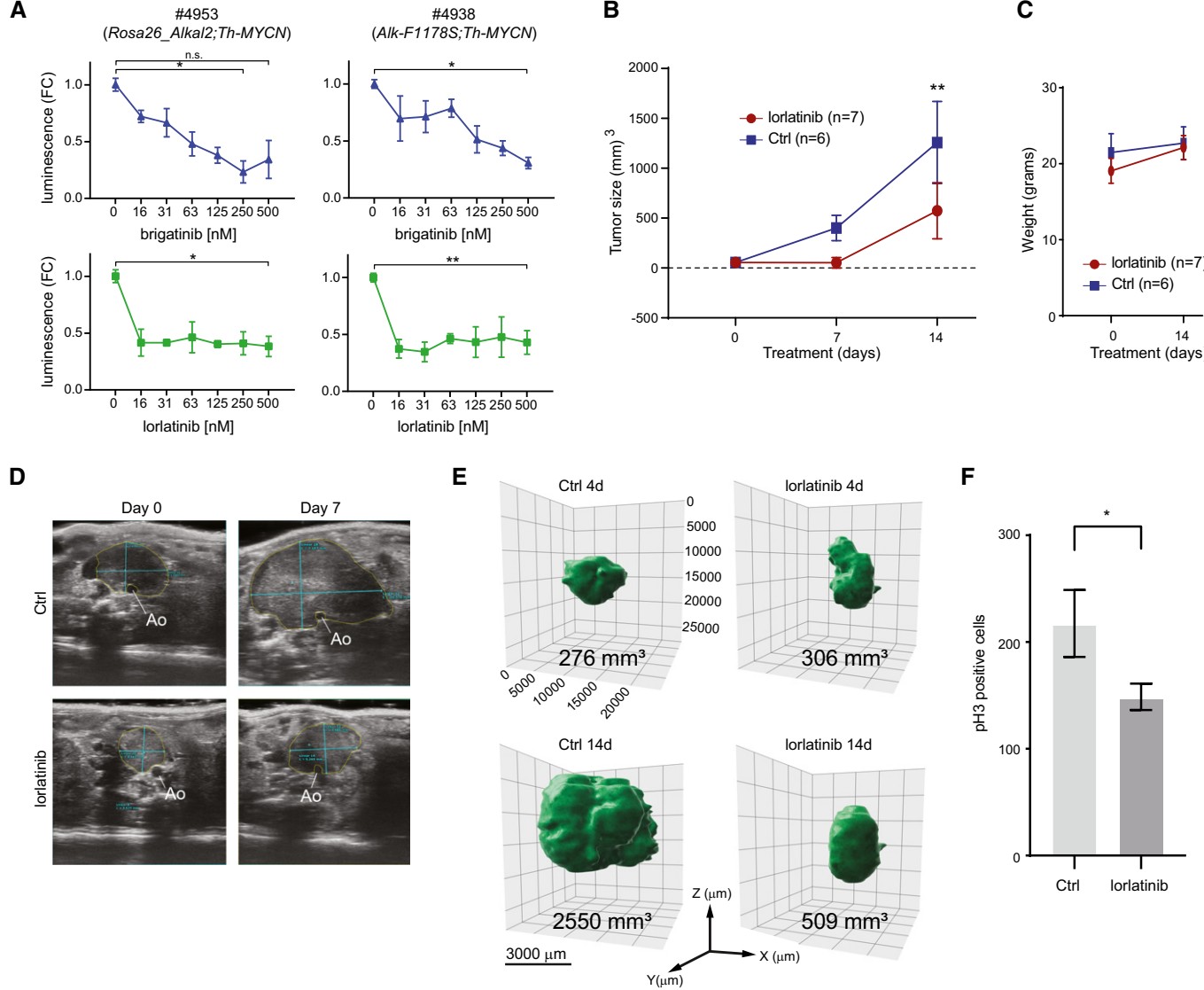

**Figure 7. ALKAL2-driven NB is sensitive to treatment with the ALK TKI lorlatinib *in vivo*.**

A Cells derived from tumours arising in *Rosa26_Alkal2;Th-MYCN* (#4953) and *Alk-F1178S;Th-MYCN* (#4938) mice are sensitive to both lorlatinib and brigatinib. The effect of increasing concentrations of each ALK TKI (as indicated) on cell confluence was analysed by IncuCyte Live Cell Analysis. Data are presented as mean ± SEM from three independent experiments. *P < 0.05, **P < 0.005; two-tailed paired Student's *t*-test.

B Tumour volume changes over time for *Rosa26_Alkal2;Th-MYCN* mice treated with lorlatinib (10 mg/kg; twice daily) or vehicle control. Tumour volume was measured by ultrasound on Days 0 and 7, and by direct measurement at Day 14. Day 0 (lorlatinib n = 7, Ctrl n = 6), Day 7 (lorlatinib n = 2, Ctrl n = 5) and Day 14 (lorlatinib n = 7, Ctrl n = 6). Data shown represent mean ± SD. **P < 0.005; two-tailed unpaired Student's *t*-test.

C *Rosa26_Alkal2;Th-MYCN* animals treated with lorlatinib did not display any significant loss of body weight compared with vehicle controls. Data shown represent mean ± SD.

D Representative ultrasound images of tumours observed in *Rosa26_Alkal2;Th-MYCN* mice with annotated measurements at Day 0 and Day 7. Tumours arise in the retroperitoneal space ventral to the aorta, Ao.

E Representative MRI imaging of *Rosa26_Alkal2;Th-MYCN* tumours in response to lorlatinib at 4 and 14 days.

F *Rosa26_Alkal2;Th-MYCN* tumours from lorlatinib or vehicle controls were analysed for phospho-histone H3 (pH3). A representative field of view for each tumour at 40× (175.740 μm²) was manually counted. Data shown represent mean ± 95% CI. P = 0.0286; Mann–Whitney test.

*Danio rerio* (Mo *et al*, 2017; Fadeev *et al*, 2018). Moreover, analysis of NB cell lines and tumour samples has highlighted expression of ALKAL2 mRNA and protein in NB (Reshetnyak *et al*, 2015; Javanmardi *et al*, 2019). Indeed, the *ALKAL2* genetic locus lies on chromosome 2p, in a region harbouring *ALKAL*, *MYCN* and *ALK* that is

often subject to chromosomal gain—so called "2p-gain"—in NB (Javanmardi *et al*, 2019).

Given the observation that "2p-gain" patients exhibit a poor prognosis within the NB patient population, our hypothesis was that *ALKAL2* dysregulation may be able to promote initiation and

progression of NB (Jeison *et al*, 2010; Javanmardi *et al*, 2019). This is supported by a number of observations over the last decade reporting that high *ALK* expression and/or activity are important for NB cell growth as well as predictive of poor prognosis in patients (Lamant *et al*, 2000; Osajima-Hakomori *et al*, 2005; Janoueix-Lerosey *et al*, 2008; Mosse *et al*, 2008; Passoni *et al*, 2009; Duijkers *et al*, 2012; Wang *et al*, 2013; Regairaz *et al*, 2016). This is further reinforced by a recent report in which 41% of NB tumour samples expressed high levels of ALK protein, which is in excess of the estimated 8–10% of primary NB that harbours an ALK mutation (Chang *et al*, 2020). It is clear from genetic and functional studies that two of the loci at 2p, *ALK* and *MYCN*, are intimately involved in the development of NB. From a mechanistic point of view, ALK regulates the expression of *MYCN*, and MYCN regulates the expression of *ALK* (Schonherr *et al*, 2012; Hasan *et al*, 2013). A third loci at 2p, *ALKAL2*, encodes for the ALKAL2 ligand that robustly stimulates ALK (Guan *et al*, 2015; Reshetnyak *et al*, 2015). Here, we show that ALKAL2 overexpression collaborates with *Th-MYCN*, driving highly aggressive and rapid onset NB, similar to that observed in *Alk-F1178S;Th-MYCN* animals. Indeed, estimated median survival for *Rosa26_Alkal2;Th-MYCN* and *Alk-F1178S;Th-MYCN* animals was similar, reached at 10 and 10.4 wks, respectively, which compares with an undefined median survival in the *Th-MYCN* animals (Fig 4F).

Our overall findings indicate a high level of NB penetrance in *Rosa26_Alkal2;Th-MYCN* animals in our survival experiment. Analysis of *Alkal2-* and *Alk-F1178S*-induced mouse tumours at the DNA level led to the detection of very few genetic alterations. Importantly, we did not observe any alterations of areas syntenic to chromosomal regions reported in human NB, such as 17q, 11q or 1p or in either *Tert* or *Atrx*. Nor did we detect any *Alk* mutations, potentially activating or otherwise, in NB arising in *Rosa26_Alkal2;Th-MYCN* animals. Previous characterization of *Th-MYCN* tumours has identified several partial and chromosomal gains and losses (Weiss *et al*, 1997; Hackett *et al*, 2003; Heukamp *et al*, 2012; Rasmuson *et al*, 2012). In genetically engineered *Alk* knock-in models, variable genetic alterations were noted dependent on genetic background, with more aggressively arising NB exhibiting less chromosomal aberrations (Heukamp *et al*, 2012; Cazes *et al*, 2014). While our identification of small focal deletions in *Tcf4* in ALKAL2-driven NB analysed is interesting, further investigation will be required to determine whether this has any functional significance. In general, the lack of widespread genetic alterations observed in either ALKAL2- or ALK-F1178S-induced NB in this study is in keeping with the highly penetrant and aggressive NB observed.

ALK TKIs are currently employed in NB, particularly in patients in which ALK mutations are identified, and a number of clinical studies are ongoing (https://clinicaltrials.gov). The first clinical study in NB employed the first-generation inhibitor crizotinib (Mosse *et al*, 2013). Since then, a range of ALK TKIs including ceritinib, lorlatinib, brigatinib, alectinib and repotrectinib have been explored in a preclinical NB setting as well as in several published clinical case reports (Heukamp *et al*, 2012; Guan *et al*, 2016; Infarinato *et al*, 2016; Iyer *et al*, 2016; Siaw *et al*, 2016; Guan *et al*, 2018; Alam *et al*, 2019; Cervantes-Madrid *et al*, 2019). While ALK mutations are identified in less than 10% of primary NB cases, this number is now appreciated to be far higher in the relapsed NB population (Schleiermacher *et al*, 2014; Eleveld *et al*, 2015).

However, we are currently unable to define the number of NB cases in which ALK signalling is activated and contributing to NB development. This is particularly relevant for NB cases that exhibit "2p-gain", in which ALKAL2, MYCN and ALK are potentially misregulated. In addition to showing that ALKAL2 collaborates with MYCN in our genetically engineered mouse models, we also provide evidence that ALK TKI treatment inhibits growth of ALKAL2-driven NB. Our experiments here have mostly employed lorlatinib, an ALK TKI that is currently used clinically. These results show for the first time that additional NB patient populations may benefit clinically from ALK-targeted therapy. This finding has important implications, since ALK TKIs appear to be generally well tolerated (Mosse *et al*, 2013; Mosse *et al*, 2017; Guan *et al*, 2018). While our focus here has been on ALK TKIs, several studies have investigated antibody based approaches that target that ALK extracellular domain, which would be interesting to test in our ALKAL2-driven NB models (Carpenter *et al*, 2012; Sano *et al*, 2019).

Our results show ALKAL2 stimulation of NB cells results in the activation of ALK downstream signalling pathways, as measured at the level of RNA and protein responses. Many of the targets identified in our study have previously been identified on addition of ALK TKIs to NB cells, such as STAT3, CRK, FOXO3 and PTPN11 (Emdal *et al*, 2018; Van den Eynden *et al*, 2018). Our datasets identify a set of early response transcription factors that are upregulated by ALKAL2 stimulation in an ALK-dependent manner, and these core transcription factors are also highly responsive to inhibition of ALK in ALK-driven NB cell lines that harbour ALK activating mutations (Van den Eynden *et al*, 2018). Our investigation of total protein levels in response to ALKAL2 also identified FOXO3 as being downregulated at the protein level in response to ALKAL2 stimulation, highlighting the complexity of regulation at both transcriptional and protein regulatory levels downstream of ALK activation. It is interesting that the ALKAL2-induced transcriptomic response observed in *Rosa26_Alkal2;Th-MYCN* is weaker than that seen in *Alk-F1178S; Th-MYCN* tumours. It is possible that the mutant ALK-F1178S receptor displays different signalling and trafficking kinetics that may result in a stronger response. While further investigation is needed to understand this better, previous work has noted abnormal trafficking of mutant ALK (Mazot, Cazes *et al*, 2012).

At the molecular level, our data identified robust upregulation of VGF in both ALKAL2-driven NB cell lines and mouse tumours, a finding also noted by Cazes and coworkers (Cazes *et al*, 2014). The *VGF* locus encodes a precursor polypeptide, which is processed to generate a complex variety of secreted products with functions that are not well understood at this time (Lewis *et al*, 2015). However, the increased levels of VGF observed in both ALKAL2- and ALK-F1178S-driven NB are of interest given a recent report that VGF expression in glioblastoma promotes tumour survival and growth (Wang *et al*, 2018). Although we have been unable to define a role for this interesting secreted molecule in this work, we show that high levels of VGF expression correlate significantly with poor prognosis in NB patient data. Thus, the role of VGF in tumorigenesis seems worthy of future more in-depth investigation in the context of NB.

Taken together, the findings presented here provide evidence of ALKAL2-driven NB that is sensitive to ALK TKI treatment. Moreover, this ALKAL2-driven NB occurs in the absence of ALK mutation. Since many neuroblastomas express ALK, our results suggest

that some "ALK mutation-negative" NB may respond to ALK TKI treatment. While "2p-gain" identifies one such chromosomal aberration in NB that may disrupt the delicate balance of ALKAL2/ALK/MYCN activity in favour of NB development, other mechanisms leading to misregulation of ALKAL2 may also exist. Currently, although mutations in the *ALKAL2 l*ocus are found in cancer sequence cohorts, these have not been explored and none have been reported as having an impact on NB development. We also note that *ALKAL2* has a CpG island in the five prime regions of its promoter that warrants exploration in future work. Our results suggest that a more careful analysis of ALK signalling activity in NB tumours in addition to ALK mutation genetic status may identify NB patients that would benefit from ALK TKI therapy.

# Materials and Methods

### Immunoblotting

All NB cell lines were cultured in RPMI 1640 or Dulbecco's modified Eagle's medium with 10% foetal bovine serum and 1% penicillin and streptomycin. Immunoblotting of cell lysates was performed as described in Van den Eynden *et al* (2018). Mouse tumour samples were lysed according to the manufacturer's instructions (Qiagen, cat. #85300). Protein concentration was measured using BCA assay (Pierce; #23225), and proteins subsequently separated on 7.5% bis-acryl-tris gels, and transferred to polyvinylidene difluoride membranes (Millipore, cat.# IPVH00010). After blocking in 5% bovine serum albumin (BSA; phosphoprotein blots) or 5% milk, and incubation with primary antibodies (detailed in Table EV5) overnight at 4°C, secondary antibodies (detailed in Table EV5) were added for 1 h at RT. Enhanced chemiluminescence substrates were used for detection (GE Healthcare, cat.# RPN2232). Lorlatinib and brigatinib were from Selleckchem.

### Stimulation of ALK with ALKAL2

NB1 cells were treated with or without 30 nM lorlatinib for 2 h prior to 30-min stimulation with 1 μg/ml of mouse ALKAL2 (CSB-YP772138MO, CUSABIO) or 1 μg/ml of ALK agonist monoclonal antibody mAb46 ((Moog-Lutz *et al*, 2005), gift from Professor Marc Vigny). Non-stimulated cells were used as experimental control. Cells were lysed in 1× SDS sample buffer and then subjected to SDS–PAGE. PC12 cells ($2 \times 10^6$ cells per electroporation) were electroporated with 1 μg of pcDNA3 empty vector or 1 μg of pME185-mALK plasmid and then seeded into collagen-coated six-well plates. After overnight culture, cells were subjected to serum starvation for 24 h prior to further treatment. Cells were treated with or without 30 nM lorlatinib for 2 h prior to 30-min stimulation with 1 μg/ml of mouse ALKAL2 (CSB-YP772138MO, CUSABIO) or 1 μg/ml of human ALKAL2. Cells were directly lysed in 1× SDS sample buffer and then subjected to SDS–PAGE.

### Tumour-derived NB cell culture and ALK TKI treatment

Tumours were harvested and dissociated cells cultured in DMEM (Gibco BRL, Life Technologies) supplemented with 10% inactivated foetal bovine serum (Gibco) and 1% penicillin/streptomycin (Gibco). Cells were seeded in 48-well plates precoated with 0.4% solution of type I bovine collagen solution (Advanced BioMatrix, lot no. 7434). The following day, cells were treated with either brigatinib (0, 16, 31, 63, 125, 250, 500 nM; Selleckchem) or lorlatinib (0, 16, 31, 63, 125, 250, 500 nM; Selleckchem). ATP content in the treated cells was determined by CellTiter-Glo 3D cell viability assay (Promega) according to the manufactures' protocol.

### Tumour-derived NB cell line immunoblotting, spheroid formation and viability

Cells were seeded into six-well plates precoated with 0.4% type I bovine collagen solution. The following day, cells were serum starved for 18 h and then treated with brigatinib (0, 150 nM) for an additional 6 h. After washing by phosphate-buffered saline (PBS), cells were lysed by lysis buffer (50 mM Tris–HCl, 100 mM DTT, 2% SDS with protease/phosphatase inhibitor cocktail [Roche]) on ice for 30 min and then centrifuged at 4°C for 25 min. Spheroid formation assays were performed by seeding $3 \times 10^5$ cells (cell line #3456) or $6 \times 10^5$ cells (cell line #3540) into six-well ultra-low attachment plates (Corning, no. 3471) prior to treatment with brigatinib (0, 150 nM; Selleckchem) for 4 days. The spheroid number was separately monitored and analysed by IncuCyte® Live Cell Analysis (Essen BioScience). Spheroids with diameter > 100 μm were counted. For spheroid viability analysis, cells were seeded into 96-well ultra-low attachment plates (Corning, no. 7007) at the indicated cell number per well for 3 days. Brigatinib (0, 150 nM) was subsequently added and re-feed every other day for another 10 days. The ATP content in the treated cells was determined by CellTiter-Glo 3D cell viability assay (Promega) according to the manufactures' protocol. For viability assays, cells were plated on 48-well plates (#3456 – 7,500 cells/per well and #3540- 2,500 cells/per well prior to treatment with brigatinib (125 and 500 nM). Cell confluence was monitored and analysed by IncuCyte® Live Cell Analysis. Also, treated cells were incubated with 55 mM resazurin (Sigma-Aldrich) for 2 h at 37°C and viability evaluated by fluorescence readout analysed on a TECAN microplate reader at the indicated time point.

### Generation of the *Alk-F1178S* knock-in mouse

*Alk-F1178S* mice were custom generated by PolyGene Transgenetics (Polygene AG, Switzerland), shown schematically in Appendix Fig S3. Residue F1178 lies within exon 23 that corresponds to nucleotides 3,528–3,657 and amino acids 1,076–1,219 (ALK sequence via NCBI NM_007439). The targeting vector contained exons 19–22 followed by a loxP site, minigene from exon 23 to the end, polyA, neo-cassette flanked by FRT sites, the second loxP and mutated version of exon 23. This construct allowed us to generate both a conditional knock-in line and non-conditional knock-in by Cre-recombinase microinjection. The neo-cassette was removed from the conditional knock-in line by crossing mice with Flp-deleter mice. In the mouse model generated here, mutation of F1178 in exon 23 results in an activated ALK-F1178S RTK under the control of physiological transcriptional regulation elements at the *Alk* locus. Animal genotypes were verified via PCR, Southern and Western blotting analyses, and the presence of the *Alk-F1178S* mutation was confirmed in the WGS analyses. Heterozygous intercrosses were performed for the analysis of homozygous mice. Mice of all three

genotypes (+/+, −/− and +/−) were obtained, with mice homozygous for the *Alk-F1178S* mutation born and viable, with expected Mendelian ratios. Since both heterozygous and homozygous *Alk-F1178S* mutant mice of both genders were fertile, a colony of *Alk-F1178S* mice was established. For genotyping the following primers: 5′-GAGAAGACTGCCTCTCACTC-3′ and 5′–CCCTTTCAGAAGCCAGTCCTT-3′ were used. Mice were backcrossed to 129X1/SvJ strain (JAX stock #000691).

### Generation of the *Rosa26_Alkal2* transgenic mouse

*Rosa26_Alkal2* mice were custom generated by Ozgene Pty Ltd (Bentley DC, Australia) through homologous recombination of codon-optimized Alkal2 cDNA into the ubiquitously expressed Gt (ROSA)26S locus (Appendix Fig S4). Cre deletion resulted in transgenic mice constitutively expressing ALKAL2 (Appendix Fig S4). *Rosa26_Alkal2* mice of Tg/Tg, Tg/T0 and T0/T0 genotypes were born at the expected Mendelian ratio and were viable and fertile. Transgenic *Rosa26_Alkal2* animals were backcrossed to 129X1/SvJ (JAX stock #000691) for at least 5 generations. *Rosa26_Alkal2* mice were genotyped on ear and/or tail at weaning. Genomic DNA was extracted employing DNeasy Blood & Tissue Kit (Qiagen Cat. # 69506) and PCR amplified using primers: primers: 5′-CGCTAAATTCTGGCCGTT-3′ and 5′- ACCAGGTTAGCCTTTAAG-3′ producing amplicons of length: 843 bp for *Rosa26_Alkal2* allele and 0 bp for *wild-type*. Insertion of *Alkal2* into the Gt(ROSA)26S locus was confirmed in WGS analyses.

### Additional mouse husbandry

*Th-MYCN* hemizygous mice (Weiss *et al*, 1997) were on genetic background 129X1/SvJ. *Th-MYCN* genotyping was performed with the following primer pair: 5′-TGGAAAGCTTCTTATTGGTAGAAACAA-3′ and 5′-AGGGATCCTTTCCGCCCCGTTCGTTTTAA-3′. All animal experiments were performed in accordance with Animal Ethics Permits (1890-2018) and (A230-2014).

### Ultrasound and MRI imaging

Ultrasound imaging of tumours was carried out using VisualSonics VEVO-770 high-frequency ultrasound system (VisualSonics) or Vevo 3100 Imaging System (VisualSonics). Acquired 3D images were used to calculate tumour volume in Vevo Lab (VisualSonics). The diameter of the tumour was noted for three dimensions approximately perpendicular to each other, and these were used to calculate the tumour volume by applying the formula of the volume for a ellipsoid ( $Volume = \frac{4}{3}\pi r_1 r_2 r_3$, where $r_1$, $r_2$ and $r_3$ are the radius of the three diameters). MRI images were obtained with a 7 Tesla Bruker BioSpin 70/20 Avance I MR system (Bruker BioSpin GmbH, Ettlingen, Germany), equipped with a maximum 400 mT/m gradient system and a 30-mm transmit/receive volume coil (RAPID Biomedical GmbH, Germany) in ParaVision 5.1 software. T2-weighted 2D RARE images were acquired with Turbo factor 8 and fat suppression, 3821.9-ms repetition time and 28.236-ms echo time. Forty axial slices, with 0.5 mm thickness and 0.7 mm interslice distance, were imaged. The resulting voxel size was $0.2000 \times 0.2007 \times 0.500$ mm$^3$. Images were obtained on day 4 and 14 after treatment start. For scanning, mice were anesthetized using isoflurane in oxygen with an induction dose of 4% and a maintenance dose of 2.5%. Respiration was monitored using a pillow-type pressure sensor (SA Instruments, Inc., NY, USA), and body temperature was maintained by a heating pad on the animal holder. For 3D volume reconstruction analysis, MRI images of tumours were processed with Imaris 7.3.0 (Bitplane, Zurich), using the "Surpass" function. Individual MRI images were manually labelled to render tumour volumes.

### Tissue preparation for *Alk-F1178S/Th-MYCN* tumour and ganglia analysis

Postnatal day 9 animals were euthanized by isoflurane and dissected trunks fixed in 4% formaldehyde in PBS for 24 h at RT. After washing in PBS, specimens were decalcified in 0.3 M EDTA for 24 h, and trunks were cut exactly along the midline of the spine and placed in embedding cassettes. Decalcification in 0.3M EDTA was continued for 24–48 more hours. After overnight washing in PBS, tissues were embedded in paraffin (Lecia TP1020, Lecia Microsystems, Wetzlar, Germany, Sakura) and cut into longitudinal 7-µm-thick sections and mounted on SuperFrost Plus slides (Menzel-Gläser, Thermo Scientific). Collected tumour tissue was fixed in 4% formaldehyde in PBS for 4–6 h in room temperature, then processed and embedded in paraffin (as described above). After deparaffinization, slides were either stained with haematoxylin and eosin for ganglia sizes and morphology assessment. Epitope retrieval was achieved either by HIER in 25 mM Tris, 1 mM EDTA pH 9.0 – for Ki67 detection. Endogenous peroxidase activity was blocked by 20-min incubation in 0.3% H$_2$O$_2$ in PBS. After blocking in 5% milk, 0.1% Triton in PBS for 1 h in room temperature, sections were incubated overnight 4°C with primary antibody (detailed in Table EV5). Primary antibody was detected with the appropriate secondary antibody either coupled to horseradish peroxidase (HRP; Thermo Scientific Cat. #31463 and #31437). Primary antibodies were detected with biotinylated secondary antibody (Vector Labs, Cat. #BA1000) and signals amplified with VECTASTAIN Elite ABC Kit (Vector Labs, Cat. #PK6100). HRP activity was detected with the chromogenic substrate ImmPACT DAB (Vector Labs, Cat. #SK-4105). Central sections through left caeliac ganglia were chosen for the ganglia morphology analysis. Areas of ganglia cross section and areas of hyperplastic regions were quantified by using ImageJ software (Schindelin *et al*, 2012). For Ki67 expression, assessment colour threshold adjustment tool of ImageJ was applied.

### Mouse tumour monitoring and TKI treatment

#### *Alk-F1178S/Th-MYCN* study
*Alk-F1178S* WT/KI mice (background strain 75% or more of 129X1/SvJ) were crossed with *TH-MYCN* TG/0. After weaning, offspring were abdominally palpated weekly to monitor tumour development. End points for Kaplan–Meier curves were set as the day at which the developing tumour was clearly observed without palpation requiring the animal to be euthanized or the sudden death of an animal. Kaplan–Meier curves were plotted and statistically analysed employing GraphPad Prism 6.0 software.

#### *Alkal2/Th-MYCN* study
*Rosa26_Alkal2* Tg/Tg mice (129X1/SvJ, N5F1) were crossed with *Alk-F1178S* KI/KI (129X1/SvJ background strain > N10) to produce

progeny that were subsequently crossed with *Th-MYCN* Tg/0. Offspring were monitored for tumour development by abdominal palpation and daily observation. End points were defined as the day at which the mouse showed symptoms requiring the mouse to be euthanized, the spontaneous death of a mouse or the end of the study (200–204 days). At end point, animals were euthanized and tumour samples preserved in 10% neutral buffered formalin solution (Sigma-Aldrich, Cat. #HT501128), liquid nitrogen and RNAlater (Invitrogen Cat. # AM7024) for subsequent analyses. Kaplan–Meier curves were generated and analysed statistically in GraphPad Prism 8.0. The overall Kaplan–Meier (Fig 4F) was produced as follows. Five wild-type control mice, six *Rosa26_Alkal2* and five *Rosa26_Alkal2; Alk-F1178S* mice were monitored for > 200 days, without developing a tumour. Overall, 58 *Alk-F1178S* animals were followed, of which 42 were terminated before 200 days without a tumour. Of 16 *Alk-F1178S* that were monitored for 200 days, no animal developed a tumour. Out of 57 *Alk-F1178S;Th-MYCN* mice, two were excluded, nine were terminated before 200 days and were therefore censored, without developing a tumour. 45 *Alk-F1178S; Th-MYCN* mice developed a tumour within 200 days and one developed a tumour after 200 days. Out of 63 *Th-MYCN* mice, four were excluded, 29 were terminated before 200 days without developing a tumour and were censored, 20 mice developed a tumour within 200 days and 10 reached 200 days without developing a tumour. Out of 50 *Rosa26_Alkal2;Th-MYCN* mice, five were excluded, 40 mice developed a tumour within 200 days, and five reached 200 days without developing a tumour.

### TKI treatment study

*Rosa26_Alkal2* Tg/0 mice (N9 or more) were crossed with *TH-MYCN* Tg/0. Offspring were screened by ultrasound 2–3 times a week from approximately 35 days old. Tumours were followed until a size of 3–6 mm in average diameter (14–113 mm$^3$), at which a 3D scan of the tumour was made and treatment started (Day 0). For 3D scanning, animals were anesthetized with isoflurane and vital parameters were observed in regard to respiration rate, ECG and body temperature. Mice were treated twice daily with 10 mg/kg lorlatinib (Selleckchem Cat. #S7536) or the equivalent volume of control by oral gavage. Treatment consisted of either lorlatinib (2% DMSO, 30% PEG 300 [Aldrich Cat. #202371]) or Control (2% DMSO [Sigma-Aldrich, Cat. #D4540], 30% PEG 300 [Sigma-Aldrich, Cat. #202371]). A mid-treatment 3D scan was performed on day 7. At 14 days, mice were euthanized and the tumour measured to obtain the greatest diameter of the tumour in three dimensions approximately perpendicular to each other. Volume calculation of the tumour was carried out as described above. Tumour samples were stored in 10% neutral buffered formalin solution (Sigma-Aldrich, Cat. #HT501128), liquid nitrogen and RNAlater™ (Invitrogen, Cat. #AM7021) for subsequent analysis. Weight was followed every second day during treatment to detect weight loss. Statistical analysis was carried out in GraphPad Prism 8.

### Tumour tissue histopathology and immunohistochemical staining—*Alkal2/Th-MYCN* and TKI treatment studies

Tumour samples were fixed for at least 2 weeks in 10% neutral buffered formalin solution (Sigma-Aldrich, HT501128). After washing in PBS, samples were dehydrated, cleared and embedded in paraffin. 4-μm-thick sections were cut and mounted on SuperFrost Plus slides (Thermo Scientific, Cat. #J1800AMNZ) after which the slides were baked for 1 h at 65°. Slides were deparaffinized and rehydrated then subjected to epitope retrieval through sub-boiling for 45 min in IHC-Tek Epitope Retrieval Steamer Set (IHCWORLD, IW-1102) containing 0.01 M citric acid, 0.05% Tween 20 (Scharlau, Cat. #TW00200250), buffer pH6. 5%. Normal goat serum in TBS-T was used for blocking at room temperature for 1 h. Sections were incubated with primary antibodies (detailed in Table EV5) were diluted in Signalstain® Antibody Diluent (CST, #8112) over night at 4°. Sections were incubated with Signalstain® Boost IHC Detection Reagent (HRP, Rabbit; CST, Cat. #8114) to detect primary antibodies. Signalstain® DAB Substrate Kit (CST, Cat. #8059) was used to detect HRP activity. After counterstaining with Mayer's haematoxylin solution (Sigma-Aldrich, Cat. #MHS1-100ML), slides were dehydrated and mounted. Digital images of sections were obtained with Hamamatsu NanoZoomer-SQ Digital slide scanner. A representative field of view at 40× (10× for H&E staining) in NanoZoomer Digital Pathology viewer was saved as a TIF-file and subsequently cropped in ImageJ (Schindelin *et al*, 2012) to 1,000 × 1,000 pixels and again saved as a TIF-file. Digital images of slides stained with pH3 were obtained as described above though the slides were afterwards subjected to manually counting. A representative field of view, for each tumour ($n = 4$ lorlatinib treated, $n = 4$ control), at 40× in NanoZoomer Digital Pathology viewer was saved as a TIF-file. Each file was subjected to manual counting applying Affinity Designer.

### Whole genome sequencing of tumours

Whole genome sequencing (WGS) was performed on DNA from eight tumours derived from the three different transgenes; *Th-MYCN* ($n = 2$), *Alk-F1178S;Th-MYCN* ($n = 3$) and *Rosa26_Alkal2;Th-MYCN* ($n = 3$). In addition, DNA extracted from spleen from two normal mice served as control for copy number and somatic variant calling. Sequencing was performed using TruSeq PCRfree library preparation (Illumina, San Diego, CA, USA) and Illumina instrumentation (Illumina) at NGI, SciLife Laboratories, Stockholm, Sweden, for a median read depth of 25× (range 20–28×) for all samples. Mapping against mouse reference genome mm10 and single nucleotide variant (SNV) calling were performed using Sentieon TNscope (Sentieon, Mountain View, CA) while CLC Genomics workbench (Qiagen, Aahus, Denmark) was used for annotation of called SNV. Structural variant (SV) calling was performed using the tool Manta (version 1.1.1) (Chen *et al*, 2016) for identification of larger structural variation (deletions, duplications, inversions and translocations). Calls from the normal DNA were supplied to filter out germline variation and artefacts. Calling and visualization of copy number variants (CNV) was done with the tool Canvas (version 1.38.0.1554) (Roller *et al*, 2016) with coverage calculated using 0,1Mb bins along the genome and normalized to average sequence coverage of DNA from two normal mice of the used strain.

Only high quality called SNV with a minimum 10% variant allele frequency and a total read coverage of 10 were kept while discarding all variants not directly affecting protein function (i.e. retaining non-synonymous and variants affecting canonical splice sites). The remaining SNVs as wells as SVs were assessed manually through the Integrative Genomics Viewer (IGV) (Thorvaldsdottir *et al*, 2013) for removal of calls due to mapping artefacts and paralogs.

## Sample preparation for RNA-Seq and proteomics analyses

Phosphoproteomics and proteomics were performed on NB1 and IMR32 cells stimulated with ALKAL2 (1 μg/ml), for 1, 6 or 24 h and treated with lorlatinib (30 nM) simultaneously. Samples were performed in duplicate. Cells were washed once with ice-cold PBS and cell pellets submitted for analysis. Protein concentration of the lysates was determined using Pierce™ BCA Protein Assay Kit (Thermo Scientific) and the Benchmark™ Plus microplate reader (Bio-Rad) with BSA solutions as standards. For RNA-Seq experiments, NB1 and IMR32 cells were stimulated with ALKAL2 (1 μg/ml), for 1, 6 and 24 h in the presence or absence of lorlatinib (30 nM) treatment. Samples were performed in triplicate. Mouse tumour samples were lysed according to the manufacturer's instruction (Qiagen, Cat. #85300). Two samples per tumour, located in the left and right of the tumour, were analysed. Three tumours were analysed for both *Alk-F1178S;Th-MYCN* and *Rosa26_Alkal2;Th-MYCN* animals, and two tumours were analysed from *Th-MYCN* mice. Total RNA was isolated using the Promega Total RNA Isolation Kit (Promega, Cat. #Z6111), and RNA concentration was measured using NanoDrop (Thermo Scientific, Cat. #ND-2000). Samples were performed in triplicate. RNA samples were sent to Novogene for analysis.

## RNA-Seq data analysis

RNA-Seq paired-end reads (read length 150 base pairs) were aligned to the human GRCh38 (human cell line data) or GRCm38 (mice data) reference genome using hisat2 (Kim *et al* 2015). The average alignment efficiency was 91.1 and 91.0% for human and mice data, respectively. Genes were annotated using GENCODE 29 (human) or M22 (mouse) (Harrow *et al*, 2012) and quantified using HTSeq (Anders *et al*, 2015). Only coding genes were used for further analysis. Differential gene expression was determined using DESeq2 (Love *et al* 2014). Genes were considered differentially expressed if their absolute $\log_2$ fold change values were above 2 at FDR-adjusted *P* values below 0.01.

To determine the amount of *Alkal2* cDNA expression in *Rosa26_Alkal2* mice, reads were mapped to the 459 nt cDNA sequence that was used for ALKAL2 overexpression. Coverage was calculated using the R *GenomicAlignments* package.

Mice differential expression heatmaps were constructed using the R *gplots* package. Hierarchical clustering was performed using the complete-linkage clustering method and Manhattan distance function. Only genes that were differentially expressed between *Th-MYCN* and either *Alk-F1178S/Th-MYCN* or *Alkal2/Th-MYCN*) were considered.

To compare the mouse tumour gene expression signatures with a set of human cancers, a principal component analysis (PCA) was performed using five RNA-Seq datasets from The Cancer Genome Atlas (breast adenocarcinoma: $n = 1,092$; colon adenocarcinoma: $n = 456$, lung adenocarcinoma: $n = 513$, renal clear cell carcinoma: $n = 530$ and glioblastoma multiforme: $n = 154$) and one from TARGET (neuroblastoma: $n = 153$, 33 *MYCN* amplified and 120 non-amplified). Human data were downloaded from https://portal.gdc.cancer.gov/. Mouse gene IDs were mapped to their human orthologs and 16,708 coding genes for which expression data were available in all datasets were selected for downstream analysis. PCA

was performed on the 6 human cancers using the R packages Facto-MineR (Lê *et al*, 2008) and factoextra based on a set of 3,209 differentially expressed genes. These genes were identified by selecting the 500 most differentially expressed genes between each human tumour pair and between *MYCN* amplified and non-amplified neuroblastoma using the R package Limma (Ritchie *et al*, 2015). Mouse gene expression data were then mapped independently using the PC coordinates.

## Proteomics Tryptic digestion and tandem mass tag labelling and LC-MS/MS analysis

The samples were digested with trypsin using the filter-aided sample preparation (FASP) method (Wisniewski *et al*, 2009). Briefly, 30 μg from each sample was reduced with 100 mM dithiothreitol at 60°C for 30 min, transferred to 30 kDa MWCO Pall Nanosep centrifugation filters (Sigma-Aldrich), washed several times with 8 M urea and once with digestion buffer prior to alkylation with 10 mM methyl methanethiosulfonate in digestion buffer for 30 min. Digestion was performed in 50 mM TEAB, 1% sodium deoxycholate (SDC) buffer at 37°C by addition of 0.3 μg Pierce MS grade Trypsin (Thermo Fisher Scientific) and incubated overnight. An additional portion of trypsin was added and incubated for another 3 h. Peptides were collected by centrifugation and labelled using tandem mass tag (TMT) 11-plex isobaric mass tagging reagents (Thermo Scientific) according to the manufacturer instructions. After pooling of the TMT set, SDC was removed by acidification with 10% TFA. The proteins were pre-fractionated into 40 fractions by basic reversed-phase chromatography (bRP-LC) using a Dionex Ultimate 3000 UPLC system (Thermo Fisher Scientific). Peptide separation was performed using a reversed-phase XBridge BEH C18 column (3.5 μm, 3.0 × 150 mm, Waters Corporation) and a linear gradient from 3 to 40% acetonitrile in 10 mM ammonium formate buffer at pH 10.00 over 17 min, followed by an increase to 90% acetonitrile over 5 min. The fractions were concatenated into 20 fractions, dried and reconstituted in 3% acetonitrile, 0.2% formic acid.

The fractions were analysed on an Orbitrap Fusion Tribrid mass spectrometer interfaced with Easy-nLC1200 liquid chromatography system (both Thermo Fisher Scientific). Peptides were trapped on an Acclaim Pepmap 100 C18 trap column (100 μm × 2 cm, particle size 5 μm, Thermo Fisher Scientific) and separated on an in-house packed analytical column (75 μm × 30 cm, particle size 3 μm, Reprosil-Pur C18, Dr. Maisch) using a linear gradient from 5 to 35% B over 75 min followed by an increase to 100% B for 5 min, and 100% B for 10 min at a flow of 300 nl/min. Solvent A was 0.2% formic acid in water, and solvent B was 80% acetonitrile, 0.2% formic acid. MS scans were performed at a resolution of 120,000, *m/z* range 380–1,380. MS/MS analysis was performed data-dependent, with top speed cycle of 3 s for the most intense doubly or multiply charged precursor ions. Most intense precursors were fragmented in MS2 by collision-induced dissociation (CID) at a collision energy of 35 with a maximum injection time of 50 ms, and detected in the ion trap followed by multinotch (simultaneous) isolation of the top 10 MS2 fragment ions, with *m/z* 400–1,400, selected for fragmentation (MS3) by higher-energy collision dissociation (HCD) at 65% and detection in the Orbitrap at a resolution of 50,000 and *m/z* range of 100–500. Precursors were isolated in the

quadrupole with an isolation window of 0.7 *m/z*, and a dynamic exclusion within 10 ppm during 60 s was used for *m/z*-values already selected for fragmentation.

## Proteomic data analysis

The data files for the TMT set were merged for identification and relative quantification using Proteome Discoverer version 2.2 (Thermo Fisher Scientific). The search was performed by matching against the Homo sapiens Swiss-Prot Database (version November 2017, Swiss Institute of Bioinformatics, Switzerland) using Mascot 2.5 (Matrix Science) with a precursor mass tolerance of 5 ppm and fragment mass tolerance of 0.6 Da. Tryptic peptides were accepted with zero missed cleavage, variable modifications of methionine oxidation and fixed cysteine alkylation; TMT-label modifications of N-terminal and lysine were selected. Percolator was used for the validation of identified proteins, and the quantified proteins were filtered at 1% FDR and grouped by sharing the same sequences to minimize redundancy. TMT reporter ions were identified in the MS3 HCD spectra with 3 mmu mass tolerance, and the TMT reporter intensity values for each sample were normalized on the total peptide amount. Only peptides unique for a given protein were considered for quantification.

Differential protein expression was determined using the R *ROTS* package (Suomi *et al*, 2017). A hyperbolic threshold with asymptotic values of abs(log$_2$ fold change) = 0.5 and $P = 0.05$ was considered to determine differential protein expression. This implied that a protein was considered differentially expressed when:

$$-\log_{10}P > -\log_{10}0.05 + \frac{0.5}{\sqrt{(\log_2 FC)^2 - 0.5^2}}.$$

## Gene set enrichment analysis

Gene set enrichment analyses (GSEA) were performed to find Gene Ontology (GO) enrichments and to predict transcription factors (TFs) responsible for the observed transcriptional responses. GSEA was performed using Fisher's exact test. Ranked GSEA was performed on the phosphoproteomics data using the R *fgsea* package (default parameters, 10,000 permutations) with ranking based on the absolute value of the ROTS statistic. When data were available for multiple phosphorylation sites, the protein/gene with the highest absolute value was selected. Gene Ontology (GO) and reactome pathway information were downloaded from the Molecular Signatures Database v6.2 (Subramanian *et al*, 2005), and transcription factor target information was derived from RegNetwork (Liu *et al*, 2015), downloaded from www.regnetworkweb.org. Network visualization of the enriched pathways was performed using the R *igraph* package (Csardi & Nepusz, 2006).

## Neuroblastoma survival data

Kaplan–Meier survival plots by for *SRF* and *VGF* gene expression were generated in the R2:genomics analysis and visualization platform (http://r2.amc.nl). Plots were generated using the KaplanScan method for both the Versteeg dataset ($n = 88$, relapse-free survival) and the Kocak dataset ($n = 649$, event-free survival).

## Ectopic expression in the *Drosophila* eye

Ectopic expression of human and mouse ALK and ALKAL2 in the adult fly eye was done with the Gal4-UAS expression system (Brand & Perrimon, 1993). Briefly, DNA encoding either mouse ALK or mouse ALKAL2 was cloned into pUAST and verified by sequencing. Transgenic flies carrying *pUAST-mAlk* or *pUAST-mAlkal2* were generated by BestGene, Inc.. Ectopic expression of either mouse or human *mAlk/hALK* and *mAlkal2/hALKAL2* transgenes was driven by *pGMR-Gal4* (Bloomington Stock Center) in the developing eye. *UAS-hALKAL2* together with *UAS-hALK* (Guan *et al*, 2015) driven by *pGMR-Gal4* was included as a positive control.

## Statistical analysis

Statistical analyses were performed with either GraphPad Prism 7/8 software or R statistical package (v3.6). Wilcoxon rank-sum tests, Student's *t*-tests, ANOVA multiple comparisons and Mantel–Cox log-rank tests (for Kaplan–Meier curves analysis) were applied as indicated in the respective sections and figure captions. For parametric tests, data were checked for normality using Shapiro-Wilk test and for equality of variance using *F* test or Browne-Forsythe test. Multiple testing corrections were performed using the Benjamini–Hochberg method (Benjamini & Hochberg, 1995).

# Data availability

The mass spectrometry proteomics data have been deposited to the ProteomeXchange Consortium via the PRIDE partner (Vizcaino *et al*, 2016) repository with the dataset identifier PXD021792 (http://www.ebi.ac.uk/pride/archive/projects/PXD021792). The RNA-Seq data have been deposited (ArrayExpress, accession no. E-MTAB-9598 (http://www.ebi.ac.uk/arrayexpress/experiments/E-MTAB-9598/; cell line data) and E-MTAB-9600 (http://www.ebi.ac.uk/arrayexpress/experiments/E-MTAB-9600/; mice data)). All other data required to evaluate the conclusions in the paper are in the paper or Supplementary Materials.

**Expanded View** for this article is available online.

## Acknowledgements

The authors thank Anne Uv for critical comments on the manuscript. The authors would like to acknowledge support of Science for Life Laboratory/Clinical Genomics, Gothenburg, for providing assistance in computational infrastructure and of the National Genomics Infrastructure (NGI)/Science for Life Laboratory, Stockholm, for performing massive parallel sequencing. Work performed at NGI has been funded by RFI/VR and Science for Life Laboratory, Sweden. Quantitative proteomics analysis was performed by Evelin Berger at the Proteomics Core Facility of Sahlgrenska Academy, University of Gothenburg. This work has been supported by grants from the Swedish Cancer Society (RHP CAN18/729; BH: CAN18/718), the Swedish Childhood Cancer Foundation (RHP:PR2015-0096; SF:NCp2015-0061 and PR2018-0099; JG TJ2016-0088 and PR2016-2011; GU: TJ2018-0056; BH: PR2017-0110), the Swedish Research Council (RHP 2015-04466; BH:2017-01324), the Swedish Foundation for Strategic Research (RB13-0204), the Göran Gustafsson Foundation (RHP2016), a Ghent University Special Research Fund Starting Grant (JVdE:

BOF.STG.2019.0073.01), Göteborgs Läkare Sällskap (MB) and the Knut and Alice Wallenberg Foundation (KAW 2018.0057).

## Author contributions

MB, DEL, BW, TM, AEW and BA performed the mouse experiments and tumour imaging, JVdE and AC carried out the bioinformatics analysis. JVdE and JK performed the R2 database analysis. GU, W-YL, T-PC, PM-G and JG carried out the *Drosophila*, cell and biochemical analyses. SF, JG and MJ performed and analysed the WGS of mouse tumours. RHP, JVdE and BH supervised the project. RHP wrote the first draft of the manuscript with input from all authors.

## Conflict of interest

The authors declare that they have no conflict of interest.

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
