## [Review Process File · The EMBO Journal]

ALK ligand ALKAL2 potentiates MYCN-driven neuroblastoma in the absence of ALK mutation

Marcus Borenäs, Ganesh Umapathy, Wei-Yun Lai, Dan E. Lind, Barbara Witek, Jikui Guan, Patricia Mendoza-Garcia, Tafheem Masudi, Arne Claeys, Tzu-Po Chuang, Abeer El Wakil, Badrul Arefin, Susanne Fransson, Jan Koster, Mathias Johansson, Jennie Gaarder, Jimmy Van den Eynden, Bengt Hallberg and Ruth H. Palmer

DOI: [10.15252/embj.2020105784](https://doi.org/10.15252/embj.2020105784)

Corresponding authors: Ruth Palmer (ruth.palmer@gu.se) , Bengt Hallberg (bengt.hallberg@gu.se), Jimmy Van den Eynden (jimmy.vandeneeynden@ugent.be)

Review Timeline:

Submission Date:	29th May 20
Editorial Decision:	30th Jun 20
Revision Received:	6th Oct 20
Editorial Decision:	19th Oct 20
Revision Received:	19th Oct 20
Accepted:	23rd Oct 20

Editor: Daniel Klimmeck

Transaction Report:

Dear Dr Palmer,

Thank you for the submission of your manuscript (EMBOJ-2020-105784) to The EMBO Journal. Please accept my apologies for the delay with the peer-review of your work due to protracted referee input and detailed discussions in the team. Your manuscript has been initially sent to three reviewers, however one referee indicated a (positive) conflict of interest only during peer-review. We have thus considered input from the other two experts, which I enclose below.

As you will see, the referees acknowledge the potential interest and novelty of your results, although they also express a number of issues that will have to be conclusively addressed before they can be supportive of publication of your manuscript in The EMBO Journal. In particular, referee #2 raises concerns regarding the human pathophysiological relevance of your findings and requires complementary experiments (ref#2, pts 1,2,6). Referee #1 agrees in that histopathological analyses should be expanded (ref#1, pts 9,11). In addition, the reviewers raise a number of points related to statistics and replicates, data presentation as well as discussion of the findings and context, which would need to be conclusively addressed to achieve the level of robustness and clarity needed for The EMBO Journal.

I judge the comments of the referees to be generally reasonable and given their overall interest, we are in principle happy to invite you to revise your manuscript experimentally to address the referees' comments.

Please let me know any time if you have additional questions or need further input on the referee comments.

Please see below for additional instructions for preparing your revised manuscript.

Thank you for the opportunity to consider your work for publication. I look forward to your revision.

Kind regards,

Daniel Klimmeck

Daniel Klimmeck, PhD
Editor
The EMBO Journal

Please also check that the title and abstract of the manuscript are brief, yet explicit, even to non-

specialists.

Before submitting your revision, primary datasets (and computer code, where appropriate) produced in this study need to be deposited in an appropriate public database (see <https://www.embopress.org/page/journal/14602075/authorguide#datadeposition>).

The accession numbers and database should be listed in a formal "Data Availability" section (placed after Materials & Method) that follows the model below (see also <https://www.embopress.org/page/journal/14602075/authorguide#availabilityofpublishedmaterial>). Please note that the Data Availability Section is restricted to new primary data that are part of this study.

Data availability

Our journal also encourages inclusion of *data citations in the reference list* to directly cite datasets that were re-used and obtained from public databases. Data citations in the article text are distinct from normal bibliographical citations and should directly link to the database records from which the data can be accessed. In the main text, data citations are formatted as follows: "Data ref: Smith et al, 2001" or "Data ref: NCBI Sequence Read Archive PRJNA342805, 2017". In the Reference list, data citations must be labeled with "[DATASET]". A data reference must provide the database name, accession number/identifiers and a resolvable link to the landing page from which the data can be accessed at the end of the reference. Further instructions are available at <https://www.embopress.org/page/journal/14602075/authorguide#referencesformat>

- a point-by-point response to the referees' comments, with a detailed description of the changes made (as a word file).
- a word file of the manuscript text.
- individual production quality figure files (one file per figure)
- a complete author checklist, which you can download from our author guidelines (<http://emboj.embopress.org/authorguide>).

- Expanded View files (replacing Supplementary Information)

Further information is available in our Guide For Authors:

The revision must be submitted online within 90 days; please click on the link below to submit the revision online before 28th Sep 2020.

Link Not Available

Referee #1:

This is a nicely written study that outlines the role of the ALK ligand ALKAL2 in driving neuroblastoma development independent of activating mutations in ALK. While it has been known for some time that mutant forms of ALK which are ligand-independent, are expressed in neuroblastoma, the effects of full-length ALK towards the pathogenesis of neuroblastoma and whether this is ligand-dependent has not been investigated. Indeed, for many years, the natural ligands for ALK were unknown and had been incorrectly attributed to other proteins. Hence, the previous discovery of ALKAL2 by this research group was an impressive step forward in our understanding of the normal physiological role of ALK. This group now follow-up on this important work to show that ALKAL2 also plays a role in the development of neuroblastoma and furthermore that ALKAL2-driven ALK activity is targetable by ALK tyrosine kinase inhibitors inducing disease remission in cell line and murine models. These data therefore represent a huge step forward in our understanding of the pathogenesis of neuroblastoma.

I have no major concerns with regards to the data that are presented. The experiments have been conducted with the most appropriate model systems and these data, as presented are sound and conclusive.

Minor concerns:

1. In some places it is not stated how many repeats and whether these are biological or technical replicates have been conducted for each experiment, e.g. Western blots.
2. Supplementary Figure S1 Western blots - I cannot see protein bands for ALK nor pALK for the IMR-32 cell lines - perhaps the contrast can be adjusted and a second image shown of the higher contrast so that these are visible.
3. spell out the meaning of the acronym SRF at first use
4. Figure 1 - Western blots could be added for ARC, EGR2 and 3 if antibodies are available. A

western blot for IMR32 could also be shown in the supplementary data.

5. Figure 2 - were any of the proteins encoded for by the transcripts detected in figure 1 also seen in the proteomics analysis?
6. Could the authors please mark on the volcano plots with lines where the cut-off lies for significance and also for the logFC
7. Figure 2e - It looks to me like the post-translational modification of FOXO3 detected does not change following treatment with lorlatinib in the volcano plot, but does in the Western blot - do the authors have an explanation for this? Can the authors also show the data for the 24 hr timepoints with lorlatinib.
8. As for the proteomics in figure 2 - do the corresponding transcript levels change in the data presented in Figure 1?
9. Figure 3E - could the authors expand on the differences in the histopathology of the MYCN versus ALK mutant/MycN tumours, particularly the description of the H&E images
10. Figure 3 legend p-values - can the authors clarify which comparisons the stated p-values relate too. The figure has *, ** or *** and the legend states an absolute value, but the value for which comparison?
11. Figure 4B - can the authors expand on the sites of tumour development - they say abdominal but attached to which organ?
12. Figure 4D - it looks like ALK levels increase in the ALKAL/Myc mouse- is ALK expressed at a higher level in these mice or is it better stabilised? A pALK blot would also be beneficial here to see if phosphorylation levels achieved with ALKAL and WT ALK are higher than those in the mutant ALK mice compared to the ThmycN (ALK WT mouse) tumours - maybe higher levels of phosphorylation might account for higher ALK expression levels due to stabilisation of the protein??
13. Fig 5G - are there any significant differences in overall survival attributable to VGF levels?
14. Figure 5F, is there a negative control? What does VGF expression look like in Th-MycN mouse tumours?
15. It is interesting that the mutant form of ALK activates more genes than ALKAL activated WT ALK (Figures 5C and D). Can the authors expand on this finding?
16. FIGURE 6D - the authors state that smaller spheroids are more sensitive to brigatinib - are these data significant? No p-values are presented.
17. Figure 7e - the legend says 5 and 15 days whilst the figure says 4 and 14.
18. Materials and methods - a lot of antibodies are described but were they all used in the manuscript?

Referee #2:

Borenäs et al. make a case for the ALK ligand, Alkal2, as a potential driver of tumorigenesis and tumor progression in neuroblastoma (NB). They provide strong experimental data, including transgenic in vivo models, to support an oncogenic role of Alkal2 overexpression. This has potential relevance in the clinic, as the authors show that Alkal2-driven NB are sensitive to ALK tyrosine kinase inhibitors. This is a compelling and very interesting story that beautifully combines tumor biology with therapeutic intervention. Generally, the data is very strong, but in the latter parts of the manuscript the level of scientific rigor is somewhat decreasing.

Strengths:

- multi-omics approach for unbiased identification of Alkal2 upregulation in NB cell lines
- novel transgenic mouse model supporting Alkal2 as driver of tumorigenesis and progression in NB

Weaknesses:

- lack of survival data for ALK TKI application in vivo
- no validation of Alkal2 deregulation is driving NB in human patients

Major Points:

- 1) The authors try to emphasize the point that aberrant ligand-receptor signaling promotes tumor formation and progression in NB, particularly in the absence of oncogenic driver mutations in ALK. This point could be substantially stronger if there is evidence for ALK phosphorylation in ALK non-mutant patient tumors, or if the authors could verify that the identified cluster of 6 downstream transcription factors are upregulated in human samples (and/or linked to patient outcome).
- 2) Better evaluation of histopathological hallmarks of NB in both Alk-F1178S/Th-MYCN and Alkal2/Th-MYCN tumors would strengthen the case that the tumors in the respective animal model closely resemble human disease pathology. This applies to Figs. 3 and 4. On a related note, do R26-Alkal2 mice show hyperplasia in celiac ganglia like Alk-F1178S mice?
- 3) Fig. 4A presents survival curves for Alk-F1178S/Th-MYCN and R26:Alkal2/Th-MYCN mice (among others). The number of mice presented for Alk-F1178S/Th-MYCN and R26:Alkal2/Th-MYCN appears fairly low (n=4 and 7, respectively). For the Alk-F1178S/Th-MYCN the authors present a larger cohort in Figure 3D, this should also be used in Figure 4A. For the R26:Alkal2/Th-MYCN the authors mention in the discussion (p.18 l.10) that 3/24 mice have not developed tumors at 200 days. I think in this light Fig. 4A is somewhat misleading, and the authors should include the full cohort of 24 R26:Alkal2/Th-MYCN animals in the Kaplan-Meier curve.
- 4) In Fig. 4D it is unclear to me why the Alkal2 band in R26:Alkal2/Th-MYCN samples is of the same intensity as the other samples. Based on the overexpression of Alkal2 in this mouse line I would have expected increased protein levels of Alkal2.
- 5) Fig. 6B/C: 'spheroid area' is a very unusual readout for sphere formation. Importantly, this does not reveal whether changes in sphere area are due to sphere formation, cell adhesion, or cell death. A more direct quantification of sphere numbers, preferably in limiting dilution format, would be better to support this point. I would further encourage the authors to include cell viability assays in Figs. 6/7, as the quantification of cell confluence may also be skewed by cell contractility or other processes not related to changes in viability.
- 6) Ideally, a survival analysis after lorlatinib treatment should be included in Fig. 7.
- 7) In several instances, statistical analysis is missing. This includes Fig. 6 panels A, C, D and Fig. 7 panel A

Minor Points:

- 1) Re-labelling Fig. 5B as Alkal2/MYCN vs MYCN would add clarity.
- 2) Fig. 5G: the survival analysis should be based on the 'median', not a 'scan' method which plots the best fit. This can force p values for data that are not necessarily meaningful.
- 3) Fig. 7B lacks a description of the statistical test that has been used.

4) Fig. S1: As the authors note, IMR-32 cells show lower expression of ALK. The ALK, p-ALK and p-AKT bands are hardly visible for IMR-32 cells. The authors should consider probing IMR-32 cells on a separate gel to show ALK expression and phosphorylation in IMR-32 cells, or perhaps showing the same membranes with longer exposure to demonstrate the same.

5) p.21 l.10 - change 'CPG island' to 'CpG island'

Manuscript: EMBOJ-2020-105784

Title: Overexpression of the ALK ligand ALKAL2 drives neuroblastoma in the absence of ALK mutation.

General response to both reviewers: We would like to thank the reviewers for their constructive comments and suggestions, and the editor for the opportunity to re-submit our manuscript. We have made a number of changes, corrections and additions that we feel have significantly strengthened the work. Despite the current difficult working circumstances we have attempted to address all of the reviewers comments, and as detailed below, we outline the changes we have made.

We have not been able to complete all of the mouse experiments asked of us by the reviewers, in part due to ethical restrictions on our treatment protocols and in part due to breeding problems in our mouse colony. Our breeding problems are a direct result of our university building a new faculty building next to the mouse house with bedrock blasting every Tuesday and Thursday, which has severely limited our mouse breeding's due to stress (they are on a sensitive 129 background). Having said this, we feel that our revised manuscript has been tightened up on many fronts and that the main message is built on a very solid foundation. We very much hope that the reviewers will also appreciate our revised manuscript.

Below is a detailed point-by-point response to the individual reviewers' comments.

Referee #1:

This is a nicely written study that outlines the role of the ALK ligand ALKAL2 in driving neuroblastoma development independent of activating mutations in ALK. While it has been known for some time that mutant forms of ALK which are ligand-independent, are expressed in neuroblastoma, the effects of full-length ALK towards the pathogenesis of neuroblastoma and whether this is ligand-dependent has not been investigated. Indeed, for many years, the natural ligands for ALK were unknown and had been incorrectly attributed to other proteins. Hence, the previous discovery of ALKAL2 by this research group was an impressive step forward in our understanding of the normal physiological role of ALK. This group now follow-up on this important work to show that ALKAL2 also plays a role in the development of neuroblastoma and furthermore that ALKAL2-driven ALK activity is targetable by ALK tyrosine kinase inhibitors inducing disease remission in cell line and murine models. These data therefore represent a huge step forward in our understanding of the pathogenesis of neuroblastoma.

I have no major concerns with regards to the data that are presented. The experiments have been conducted with the most appropriate model systems and these data, as presented are sound and conclusive.

Minor concerns:

1. In some places it is not stated how many repeats and whether these are biological or technical replicates have been conducted for each experiment, e.g. Western blots.

Authors reply: The reviewer has a good point and we have now stated the number of repeats and whether they are biological or technical throughout.

2. Supplementary Figure S1 Western blots - I cannot see protein bands for ALK nor pALK for the IMR-32 cell lines - perhaps the contrast can be adjusted and a second image shown of the higher contrast so that these are visible.

Authors reply: We have done our best to improve these blots, including a second image as suggested by the reviewer. However, ALK and pALK are always difficult when running IMR-32 together with NB1 samples due to the low levels of ALK.

3. spell out the meaning of the acronym SRF at first use

Authors reply: We thank the reviewer for spotting this; we have now fixed this in the revised version.

4. Figure 1 - Western blots could be added for ARC, EGR2 and 3 if antibodies are available. A western blot for IMR32 could also be shown in the supplementary data.

Authors reply: We have tried very hard to look at ARC and EGR2/3, however, we have not been able to find antibodies that work for these proteins – at least not at the endogenous level in neuroblastoma cells. During the revision process, we purchased a number of antibodies that should detect these proteins, but none detected endogenous proteins of the correct size and we were unable to generate any trustable data despite intensive effort. Therefore, we are not able to provide any solid immunoblotting data to address this point.

However, as suggested by the reviewer, we have complemented the NB1 analysis with analysis in IMR-32 cells, which is now included as Expanded View Figure 2.

5. Figure 2 - were any of the proteins encoded for by the transcripts detected in figure 1 also seen in the proteomics analysis?

Authors reply: The reviewer raises a good point here. We have checked our RNAseq data with the proteomics and phosphoproteomics data, but it is clear that we do not detect scarce proteins. In summary, for EGR1 -3, ARC, FOS, FOSB and SRF there is no data in our proteomics or phosphoproteomics and we only see them by RNASeq. However, e.g. EGR1 and FOS are clearly detected with antibodies. Since we do not detect the proteins, either by mass spectrometry or immunoblotting, we are currently unable to add any insightful data regarding this question. While we have a rich proteomics and phosphoproteomics dataset, we have been following current discussion regarding overlap and integration of proteomic (total and phosphor) and RNASeq data and realise that this is not trivial. We are currently investigating this and working to produce deeper phosphor datasets with which to tackle these challenges.

6. Could the authors please mark on the volcano plots with lines where the cut-off lies for significance and also for the logFC

Authors reply: We opted for a hyperbolic threshold in the proteomics data analysis, analogous to other proteomics tools and studies (e.g. *Singh S, proteomics 2016; Tyanova S, Nature Biotech 2016; Emdal, Science signaling 2018*). This approach implies a smoother transition between significance (log P values) and logFC cut-offs (i.e. a protein which is on the edge of being significant will be required to have a higher logFC value and vice versa). To clarify this for the reader we have added the following sentence to the figure caption: *"Dashed lines indicate differential expression thresholds (hyperbolic cut-off with asymptotes $P=0.05$ and $\log_2(FC)=\pm 2$; see Methods for details)."*

7. Figure 2e - It looks to me like the post-translational modification of FOXO3 detected does not change following treatment with lorlatinib in the volcano plot, but does in the Western blot - do the authors have an explanation for this? Can the authors also show the data for the 24 hr timepoints with lorlatinib.

Authors reply: We agree with the reviewer's interpretation. While the blots indeed suggest lorlatinib sensitivity, in contrast to the other genes we labelled in fig. 2E, FOXO3 phosphorylation at S253 does not substantially change after lorlatinib inhibition of ALK, suggesting regulation that is more complex. This complexity is further supported by the ALK-dependent transient nature of the response (disappears after 24hr in control conditions, but not after lorlatinib treatment). To illustrate this, we have now added the 24hr lorlatinib volcano plot to panel E as well as an additional panel G focusing on FOXO3 (phosphor)protein dynamics. We note that there are many examples of complex regulation in our dataset, which can be compared with e.g. STAT3 dynamics, which 'behave' rather as expected (Fig 2I). Some of these, such as FOXO3 are rather interesting, and shed light on additional layers of regulation other than phosphorylation. Despite continuing experimental analysis we do not yet fully understand what is going on in many of these cases, and therefore in the revised manuscript we have tried to highlight these observations more

clearly and hope that other investigators as well as ourselves in the future will dig into more detail to provide mechanistic insight.

8. As for the proteomics in figure 2 - do the corresponding transcript levels change in the data presented in Figure 1?

Authors reply: We have looked at a quite a few proteins, including VGF, TNC, IGFBP5 and FOSL2, and their corresponding transcript levels, and find that this becomes very complex in each individual case. Right now, we think that it is difficult to add anything useful regarding this to the manuscript without getting into a lot of detail that can become rather confusing.

9. Figure 3E - could the authors expand on the differences in the histopathology of the MYCN versus ALK mutant/MycN tumours, particularly the description of the H&E images

Authors reply: We have revised our text and included more description of the images, this can be found on page 11.

10. Figure 3 legend p-values - can the authors clarify which comparisons the stated p-values relate too. The figure has *, ** or *** and the legend states an absolute value, but the value for which comparison?

Authors reply: We have revised the legend for Figure 3 to clarify this for the reader.

11. Figure 4B - can the authors expand on the sites of tumour development - they say abdominal but attached to which organ?

Authors reply: We have now included data regarding the sites of tumor development. Interestingly, in almost all cases there is no involvement of the adrenal gland, rather the tumour arises in close proximity, most likely in the nearby peripheral nervous system. We observe this in all genotypes, and have now included this. This is rather interesting and something that we would like to follow up in future work in more detail. We have added this data as Fig 4C and have included text addressing this on pages 12/13 in the revised manuscript.

12. Figure 4D - it looks like ALK levels increase in the ALKAL/Myc mouse- is ALK expressed at a higher level in these mice or is it better stabilised? A pALK blot would also be beneficial here to see if phosphorylation levels achieved with ALKAL and WT ALK are higher than those in the mutant ALK mice compared to the ThmycN (ALK WT mouse) tumours - maybe higher levels of phosphorylation might account for higher ALK expression levels due to stabilisation of the protein??

Authors reply: We agree with the reviewer that the blot might suggest higher ALK expression in *Rosa26_Alkal2;Th-MYCN* mice. We verified this at the transcriptome level in our RNA-Seq data (Table S4) and we could not confirm any significant differential expression between *Rosa26_Alkal2;Th-MYCN* and *Th-MYCN* (logFC -0.19, $P=0.52$) nor between *Alk-F1178S;Th-MYCN* and *Th-MYCN* (logFC=0.44, $P=0.14$). Based on this, we have rerun these samples and have included another ALK panel here. Despite several attempts, we have been

unable to see a clear pALK signal in these tumour samples and have therefore not included pALK data.

13. Fig 5G - are there any significant differences in overall survival attributable to VGF levels?

Authors reply: Based on the comments from both reviewer #1 and 2 we have now updated the survival analyses to use median values as cut-offs between low/high expression. We also checked overall survival and found that this was also significant (log-rank test $P=3.3e-16$ for the Kocak cohort and log-rank test $P=5.2e-05$ for the Versteeg cohort). While overall survival is not shown in the revised Figure, we have included this information in the text (on page 15).

14. Figure 5F, is there a negative control? What does VGF expression look like in Th-MycN mouse tumours?

Authors reply: We agree with the reviewer that this would be good to include. We find that there is VGF protein in the *MYCN* tumors, however, the staining is not as strong as in the *Rosa26_Alkal2;Th-MYCN* and the *Alk-F1178S;Th-MYCN* tumours, in agreement with our RNAseq data. These data have now been included in Fig 5G of the revised version.

15. It is interesting that the mutant form of ALK activates more genes than ALKAL activated WT ALK (Figures 5C and D). Can the authors expand on this finding?

Authors reply: The reviewer is quick to notice this point, and we have also been considering what this means, and have discussed this at length in the lab. We have some ideas here about what is going on here. In our experiments in both neuroblastoma cell lines and in the model organism *Drosophila melanogaster*, we have noted that overexpression of ligand appears to downregulate the ALK protein. Moreover, when we mutate ALK, the protein appears to have a slightly different localisation, and to be more persistent. There is some previous data to support this, such as Mazot et al, *Oncogene* 2011, in which activation of ALK was shown to affect receptor trafficking. We suspect that differences in the trafficking of the wild-type versus the mutant ALK receptor may at least be part of the explanation for our observations. We have now added text in the discussion addressing this.

16. FIGURE 6D - the authors state that smaller spheroids are more sensitive to brigatinib - are these data significant? No p-values are presented.

Authors reply: We thank the reviewer for spotting this; somehow, we missed these and have now included them in the revised version. The reviewer is indeed correct that the smaller *Rosa26_Alkal2;Th-MYCN* spheroids (2000 cells or 4000 cells) are more sensitive to brigatinib treatment than the bigger spheroids (8000 cells), and this is statistically significant (p-values of 0.01 and 0.014). However, this trend was not detected in *Alk-F1178S;Th-MYCN* spheroids. While interesting, we currently have no explanation for this observation. However, we have now commented on this in the text and added the p-values in the revised manuscript.

17. Figure 7e - the legend says 5 and 15 days whilst the figure says 4 and 14.

Authors reply: We thank the reviewer for spotting this error, we have now corrected this in the revised version.

18. Materials and methods - a lot of antibodies are described but were they all used in the manuscript?

Authors reply: We thank the reviewer for pointing this out. We have now gone through the materials and methods and checked the antibodies used carefully. The antibodies and dilutions used are now clearly detailed in a table (S5) in the revised version.

Referee #2:

Borenäs et al. make a case for the ALK ligand, Alkal2, as a potential driver of tumorigenesis and tumor progression in neuroblastoma (NB). They provide strong experimental data, including transgenic in vivo models, to support an oncogenic role of Alkal2 overexpression. This has potential relevance in the clinic, as the authors show that Alkal2-driven NB are sensitive to ALK tyrosine kinase inhibitors. This is a compelling and very interesting story that beautifully combines tumor biology with therapeutic intervention. Generally, the data is very strong, but in the latter parts of the manuscript the level of scientific rigor is somewhat decreasing.

Strengths:

- multi-omics approach for unbiased identification of Alkal2 upregulation in NB cell lines
- novel transgenic mouse model supporting Alkal2 as driver of tumorigenesis and progression in NB

Weaknesses:

- lack of survival data for ALK TKI application in vivo
- no validation of Alkal2 deregulation is driving NB in human patients

Major Points:

1) The authors try to emphasize the point that aberrant ligand-receptor signaling promotes tumor formation and progression in NB, particularly in the absence of oncogenic driver mutations in ALK. This point could be substantially stronger if there is evidence for ALK phosphorylation in ALK non-mutant patient tumors, or if the authors could verify that the identified cluster of 6 downstream transcription factors are upregulated in human samples (and/or linked to patient outcome).

Authors reply: We appreciate the reviewers comment here. On looking back at the initial submitted manuscript, we realise that we have not provided much background on the evidence regarding ALK levels and activity in NB, outside of the ALK mutant status. To clarify this we now discuss results from others that provide evidence for correlation of ALK expression/activity and prognosis at the transcript and protein level, including a recently published study. These findings fit with our own unpublished data from a limited number of patient samples. However, we are not a clinical laboratory, and have only very limited access to patient samples for analysis by either immunohistochemistry or proteomics methods. We hope to be able to look at ALKAL2 expression in patient samples in a future collaboration.

As suggested by the reviewer we have looked into the highlighted transcription factors in the context of patient outcome and for EGR1, EGR2, ARC, FOS, FOSB and FOSL2 the data is very mixed depending on which dataset one looks at. We also looked at SRF, and here we saw a consistent association in all datasets looked at, where high SRF is always associated with poor outcome. We

thank the reviewer for this suggestion and have now included this data as Extended View Figure 3, where we show by both median and scan outcome related to SRF levels.

2) Better evaluation of histopathological hallmarks of NB in both Alk-F1178S/Th-MYCN and Alkal2/Th-MYCN tumors would strengthen the case that the tumors in the respective animal model closely resemble human disease pathology. This applies to Figs. 3 and 4. On a related note, do R26-Alkal2 mice show hyperplasia in celiac ganglia like Alk-F1178S mice?

Authors reply: Regarding the potential hyperplasia in R26-Alkal2 ganglia, our initial experiments suggest they are enlarged, however we have not repeated this with sufficient numbers of mice and controls to include this data. Given our current breeding speed I estimate this would take us quite a few more months to achieve. We have tried to improve the text describing the tumours arising in both Alk-F1178S/Th-MYCN and Alkal2/Th-MYCN tumors in the revised version of the manuscript. Moreover, this comment from the reviewer prompted us to compare the Th-MYCN, Alk-F1178S/Th-MYCN and Alkal2/Th-MYCN mouse tumour gene expression signatures with a set of human cancers. This was done by principal component analysis (PCA), using 5 RNA-Seq datasets The Cancer Genome Atlas (breast adenocarcinoma: n=1092; colon adenocarcinoma: n=456, lung adenocarcinoma: n=513, renal clear cell carcinoma: n=530 and glioblastoma multiforme: n= 154) and 1 from TARGET (neuroblastoma: n=153). Although this turned out to be a challenging computational task, we were very happy to see that the Th-MYCN, Alk-F1178S/Th-MYCN and Alkal2/Th-MYCN mouse tumour showed the highest similarity with human NB tumours. This new data is included as Fig 5B in the revised version of the MS.

3) Fig. 4A presents survival curves for Alk-F1178S/Th-MYCN and R26:Alkal2/Th-MYCN mice (among others). The number of mice presented for Alk-F1178S/Th-MYCN and R26:Alkal2/Th-MYCN appears fairly low (n=4 and 7, respectively). For the Alk-F1178S/Th-MYCN the authors present a larger cohort in Figure 3D, this should also be used in Figure 4A. For the R26:Alkal2/Th-MYCN the authors mention in the discussion (p.18 l.10) that 3/24 mice have not developed tumors at 200 days. I think in this light Fig. 4A is somewhat misleading, and the authors should include the full cohort of 24 R26:Alkal2/Th-MYCN animals in the Kaplan-Meier curve.

Authors reply: We have now included all data from our mouse colony (Fig 4F). We note that we also have seen one *Alk-F1178S/Th-MYCN* animal that did not present with tumor before 200 days, which is also included in this data. This data includes all mice in our colony that were monitored for 200 days as well as those mice that were used in treatment upon tumor detection. For animals that were used in treatment schedules we estimated survival as 2 weeks after tumor detection, as estimate of survival in the absence of treatment. This data provides a more accurate overview of the overall penetrance. In our initial experiment (Fig 4A), all mice were carefully monitored and controlled and we present a highly controlled experiment. We have not tried to mislead the reader, in fact we have striven to be very honest. We also cannot say whether mice that were tumour free at 200 days would have presented tumors later, however. We now have 45 *R26:Alkal2/Th-MYCN* mice, of which 5 mice have reached 200 days without presenting with a tumour, therefore we are starting to feel confident that somewhere around 10% of mice will not present with a tumour by 200 days. This makes our model highly penetrant, although not quite as much as our *Alk-F1178S/Th-MYCN*, which is almost fully penetrant at 200 days (97.8%).

4) In Fig. 4D it is unclear to me why the Alkal2 band in R26:Alkal2/Th-MYCN samples is of the same intensity as the other samples. Based on the overexpression of Alkal2 in this mouse line I would have expected increased protein levels of Alkal2.

Authors reply: We realise that this may appear confusing in the original version of the manuscript. We have been able to detect ALKAL2 protein in both human neuroblastoma cell lines (Javanmardi, 2019) and patient tumors (unpublished). This is in agreement with our RNASeq data, which detects *Alkal2* mRNA in tumor samples of all three mouse genotypes that we have used here (Table S4). As the *Alkal2* transgene is codon optimized, we can clearly state that all tumors transcribe *Alkal2* from the endogenous locus. We are unable to comment on whether the Alkal2 protein observed in the mouse tumors originates from the tumor cells themselves or from stromal tissue that contributes to the tumour, or indeed even more remotely, and this is something we are interested in looking at in future experiments. However, when we isolate cell lines from the mouse tumors, we observe a clear increase in Alkal2 protein in the cells derived from *R26:Alkal2/Th-MYCN* mouse tumors as compared with cells isolated from *Alk-F1178S/Th-MYCN* tumors. This data is now included in the revised version of Figure 6 (Figure 6B) and we have also added text in the results that hopefully clarifies this for the reviewer and for the reader.

5) Fig. 6B/C: 'spheroid area' is a very unusual readout for sphere formation. Importantly, this does not reveal whether changes in sphere area are due to sphere formation, cell adhesion, or cell death. A more direct quantification of sphere numbers, preferably in limiting dilution format, would be better to support this point. I would further encourage the authors to include cell viability assays in Figs. 6/7, as the quantification of cell confluence may also be skewed by cell contractility or other processes not related to changes in viability.

Authors reply: We appreciate the reviewer's comments here and as suggested, we have re-analyzed the data as sphere numbers instead of spheroid area. For spheroid number quantification, spheroids with a diameter larger than 100um were counted. Brigatinib treatment significantly decreased the spheroid number in both #3540 and #3546 cell lines. We have also performed cell viability assays with resazurin, and these new data are included in the revised Figure 6 (as Figure 6F).

6) Ideally, a survival analysis after lorlatinib treatment should be included in Fig. 7.

Authors reply: We appreciate the reviewer's comments here; however we have been unable to provide this. We thank the reviewer however; as this comment has meant that we have included this possibility as an option in our updated Ethical permission, which is currently under evaluation.

7) In several instances, statistical analysis is missing. This includes Fig. 6 panels A, C, D and Fig. 7 panel A

Authors reply: We thank the reviewer for spotting this; somehow, we missed these and have now included them in the revised version.

Minor Points:

1) Re-labelling Fig. 5B as Alkal2/MYCN vs MYCN would add clarity.

Authors reply: On reflection, we agree this abbreviation was confusing and thank the reviewer for the suggestion. We now use the complete names (*Th-MYCN*, *Alkal2* and *Th-MYCN, ALK^{F1178S}*) consistently in main text and figures.

2) Fig. 5G: the survival analysis should be based on the 'median', not a 'scan' method which plots the best fit. This can force p values for data that are not necessarily meaningful.

Authors reply: We have now updated the survival analyses to use median values as cut-offs between low/high expression.

3) Fig. 7B lacks a description of the statistical test that has been used.

Authors reply: We have now included this in the revised version.

4) Fig. S1: As the authors note, IMR-32 cells show lower expression of ALK. The ALK, p-ALK and p-AKT bands are hardly visible for IMR-32 cells. The authors should consider probing IMR-32 cells on a separate gel to show ALK expression and phosphorylation in IMR-32 cells, or perhaps showing the same membranes with longer exposure to demonstrate the same.

Authors reply: We have done our best to improve these blots, including a second image as suggested by the reviewer. However, ALK and pALK are always difficult when running IMR-32 together with NB1 samples due to the low levels of ALK. We have rather tried to be honest about our starting samples, IMR-32 and NB1 both respond, but a decent phosphor dataset from IMR-32 will likely require a more stringent experimental approach, with significantly more starting material and using anti-PY/anti-PS/T to immunoprecipitate targets. This is something we are considering doing in the future, but this experimental approach is unfortunately very expensive.

5) p.21 l.10 - change 'CPG island' to 'CpG island'

Authors reply: We thank the reviewer for spotting this error; we have now corrected this in the revised version.

Dear Dr Palmer,

Thank you for submitting your amended manuscript (EMBOJ-2020-105784R) to The EMBO Journal. Your revised study was sent back to the referees #1 and #2, and we have received comments from both of them, which I enclose below. As you will see the referees find that their concerns have been sufficiently addressed and they are now broadly in favour of publication.

Thus, we are pleased to inform you that your manuscript has been accepted in principle for publication in The EMBO Journal, pending the a number of minor aspects related to formatting and data representation as detailed below are addressed at resubmission.

Further, I will share additional edits and comments from our production team during the next days to be considered.

Please contact me at any time if you have further questions related to below points.

Thank you for giving us the chance to consider your manuscript for The EMBO Journal. I look forward to your final revision.

Again, please contact me at any time if you need any help or have further questions.

Kind regards,

Daniel Klimmeck

Daniel Klimmeck PhD
Editor
The EMBO Journal

>> Please introduce the maximally five keywords for your study.

>> Recheck callouts for figures 4E in the main text.

>> Please specify distinct author contributions for T.M., T.P.C., J.K., M.J., J.G. .

>> Introduce ORCID IDs for all corresponding authors (J.vdE., B.H.) via our online manuscript system. Please see below for additional information.

>> Please add a ToC on the first page of the Appendix, as well as appendix figure legends.

>> Please add the following funding information in our online manuscript system: CAN18/718, NCp2015-0061, PR2018-0099, T J2016-0088, 9PR2016-2011, T J2018-0056, 2015-04466.

>> Please rename the current 'Data and materials availability' section to 'Data availability'. Remove the web link.

>> Dataset EV Legends: There are 5 suppl. tables uploaded separately. They should be renamed "Table EV1" etc. and their respective legends should be added to the files, as a separate sheet.

>> Please rename the current 'Competing interests' statement to 'Conflict of interest'.

Please note that as of January 2016, our new EMBO Press policy asks for corresponding authors to link to their ORCID iDs. You can read about the change under "Authorship Guidelines" in the Guide to Authors here: <http://emboj.embopress.org/authorguide>

In order to link your ORCID iD to your account in our manuscript tracking system, please do the following:

1. Click the 'Modify Profile' link at the bottom of your homepage in our system.
2. On the next page you will see a box half-way down the page titled ORCID*. Below this box is red text reading 'To Register/Link to ORCID, click here'. Please follow that link: you will be taken to ORCID where you can log in to your account (or create an account if you don't have one)
3. You will then be asked to authorise Wiley to access your ORCID information. Once you have approved the linking, you will be brought back to our manuscript system.

We regret that we cannot do this linking on your behalf for security reasons. We also cannot add your ORCID iD number manually to our system because there is no way for us to authenticate this iD number with ORCID.

Thank you very much in advance.

- a point-by-point response to the referees' comments, with a detailed description of the changes made (as a word file).

- a word file of the manuscript text.

- individual production quality figure files (one file per figure)

- a complete author checklist, which you can download from our author guidelines (<https://www.embopress.org/page/journal/14602075/authorguide>).

- Expanded View files (replacing Supplementary Information)

Further information is available in our Guide For Authors:

The revision must be submitted online within 90 days; please click on the link below to submit the revision online before 17th Jan 2021.

Link Not Available

Referee #1:

I am content that the authors have addressed all of my concerns regarding the submitted manuscript. I am grateful that they have taken the considerable effort to conduct extra experiments considering the difficult situation we all find ourselves in. This research addresses an important question regarding the pathogenesis of ALK-expressing neuroblastoma and I fully recommend it be accepted in its current, revised form

Referee #2:

I commend the authors on their attempts to address all of my comments. I appreciate the difficulties surrounding some of the suggestions I made and I am impressed with the revised version of the manuscript.

The authors performed the requested changes.

Dear Dr Palmer,

Thank you for submitting the revised version of your manuscript. I have now evaluated your amended manuscript and concluded that the remaining minor concerns have been sufficiently addressed.

Thus, I am pleased to inform you that your manuscript has been accepted for publication in the EMBO Journal.

Please note that it is EMBO Journal policy for the transcript of the editorial process (containing referee reports and your response letter) to be published as an online supplement to each paper.

Also in case you might NOT want the transparent process file published at all, you will also need to inform us via email immediately. More information is available here:

http://emboj.embopress.org/about#Transparent_Process

Please note that in order to be able to start the production process, our publisher will need and contact you regarding the following forms:

- PAGE CHARGE AUTHORISATION (For Articles and Resources)

[http://onlinelibrary.wiley.com/journal/10.1002/\(ISSN\)1460-2075/homepage/tej_apc.pdf](http://onlinelibrary.wiley.com/journal/10.1002/(ISSN)1460-2075/homepage/tej_apc.pdf)

- LICENCE TO PUBLISH (for non-Open Access)

Your article cannot be published until the publisher has received the appropriate signed license agreement. Once your article has been received by Wiley for production you will receive an email from Wiley's Author Services system, which will ask you to log in and will present them with the appropriate license for completion.

- LICENCE TO PUBLISH for OPEN ACCESS papers

Authors of accepted peer-reviewed original research articles may choose to pay a fee in order for their published article to be made freely accessible to all online immediately upon publication. The EMBO Open fee is fixed at \$5,200 (+ VAT where applicable).

We offer two licenses for Open Access papers, CC-BY and CC-BY-NC-ND.

For more information on these licenses, please visit: <http://creativecommons.org/licenses/by/3.0/> and http://creativecommons.org/licenses/by-nc-nd/3.0/deed.en_US

- PAYMENT FOR OPEN ACCESS papers

You also need to complete our payment system for Open Access articles. Please follow this link and select EMBO Journal from the drop down list and then complete the payment process:

https://authorservices.wiley.com/bauthor/onlineopen_order.asp

Should you be planning a Press Release on your article, please get in contact with embojournal@wiley.com as early as possible, in order to coordinate publication and release dates.

On a different note, I would like to alert you that EMBO Press is currently developing a new format for a video-synopsis of work published with us, which essentially is a short, author-generated film explaining the core findings in hand drawings, and, as we believe, can be very useful to increase visibility of the work.

Please see the following links for a representative examples:

http://embopress.org/video_EMBOJ-2014-90147

If you have any questions, please do not hesitate to call or email the Editorial Office.

Kind regards,

Daniel Klimmeck

Daniel Klimmeck, PhD
Editor
The EMBO Journal
EMBO
Postfach 1022-40
Meyerhofstrasse 1
D-69117 Heidelberg
contact@embojournal.org
Submit at: <http://emboj.msubmit.net>

Corresponding Author Name: Ruth Palmer

Journal Submitted to: EMBO J

Manuscript Number: EMBOJ-2020-105784